# Olanzapine-induced metabolic syndrome is partially mediated by oxytocinergic system dysfunction in female Sprague-Dawley rats

Elsie D. Oduor[1]*, Peter W. Mwangi[1], Boniface M. Chege[1,2], Sharon F. Olago[1], Frederick Bukachi[1]

1 Department of Human Anatomy and Medical Physiology, University of Nairobi, Nairobi, Kenya, 2 School of Health Sciences, Dedan Kimathi University of Technology, Nyeri, Kenya

☯ These authors contributed equally to this work.
* elsieoduor@gmail.com

## Abstract

### Introduction

Olanzapine (OLZ), a second-generation antipsychotic, is associated with the development of metabolic syndrome with unclear underlying pathophysiologic mechanisms. Oxytocin (OT) influences feeding, lipid, and glucose metabolism. This study investigates whether dysfunction in the oxytocinergic system contributes to the development of olanzapine-induced metabolic syndrome.

### Methods

Twenty five (25) female Sprague-Dawley rats were housed under standard conditions and studied over 12 weeks. During the first 6-week induction phase, rats were randomized into 3 groups: normal control (vehicle treatment; normal saline; n=5), low dose (4 mg/kg olanzapine [OLZ]; n=5), and high dose (8 mg/kg OLZ; n=15). In the last 6-week treatment phase, the high dose group was re-randomized into 3 groups: negative control (8 mg/kg OLZ; n=5), positive control (8 mg/kg OLZ+500 mg/kg metformin; n=5), and test group (8 mg/kg OLZ+1 mg/kg oxytocin [OT]; n=5). The normal control and low dose groups continued unchanged. Body weight, food intake, glucose levels, OGTT, lipid profile, visceral fat, hepatic index, hepatic triglycerides, and steatosis were assessed.

### Results

At induction end, high-dose OLZ increased food intake (179±5 g), body weight (239±3 g), blood glucose (7.8±0.3 mmol/L), and impaired glucose tolerance (846±25 mmol/L·min) compared to controls (p<0.0001). Post-treatment, the test group displayed reduced food intake (163±2 g vs. 197±6 g), body weight (297±2 g vs. 376±6 g), blood glucose (5.8±0.3 mmol/L vs. 9.8±0.2 mmol/L), and improved glucose tolerance (711±14 vs. 853±9 mmol/L·min) compared to negative controls

**Data availability statement:** All relevant data are within the paper and its Supporting Information files.

**Funding:** The author(s) received no specific funding for this work.

**Competing interests:** The authors have declared that no competing interests exist.

(p < 0.0001). LDL-C, total cholesterol, serum and hepatic triglycerides, visceral adipose, and hepatic mass and steatosis were also significantly decreased in the test group compared to negative control group (p < 0.01).

## Conclusion

OLZ-induced metabolic abnormalities were mitigated by oxytocin, indicating that the oxytocinergic system hypofunction may be implicated in its pathophysiology. These results highlight OT's therapeutic potential and call for further clinical research to explore its role in the management of antipsychotic-induced metabolic syndrome.

## Introduction

Olanzapine (OLZ) is an atypical antipsychotic drug used in the management of psychiatric disorders such as schizophrenia, depression, and bipolar disorder [1]. Long-term OLZ therapy is associated with metabolic syndrome as a side effect [2,3], a complex of metabolic dysregulations including central obesity, dyslipidemia, hyperglycemia, and hypertension [3]. Metabolic syndrome is linked to cardiovascular disease, type 2 diabetes mellitus, and overall increased mortality [4]. However, the pathophysiologic mechanisms underlying olanzapine-induced metabolic dysregulation remain unclear. Proposed contributing factors include the disruption of appetite-regulating centers, reduced peripheral insulin sensitivity, and altered lipid metabolism [5].

Beyond its well-established role in reproduction, oxytocin has been found to influence feeding behavior and energy metabolism, with regard to glucose and lipid metabolism [6]. Oxytocin receptors (OXTR) are present in several organs involved in energy metabolism and utilization, e.g., adipose tissue, pancreas, liver, skeletal muscle, and the hypothalamus [7]. A deficiency in oxytocin signaling is associated with insulin resistance, weight gain, and dyslipidemia [8]. Unlike other known intervention methods like metformin, which mainly improve insulin sensitivity and decrease hepatic glucose production [9], oxytocin has been shown to regulate metabolism through both central mechanisms, such as appetite and energy balance, and peripheral actions in metabolic organs [7]. However, its involvement in olanzapine-induced metabolic syndrome is yet to be fully elucidated.

This study aimed to explore the involvement of the oxytocinergic system as a possible pathophysiologic mechanism underlying OLZ-induced metabolic syndrome in a rodent model by, analyzing dose-dependent effects (OLZ), characterizing its biochemical features, and investigating OT's therapeutic potential. We hypothesized that OLZ-induced metabolic syndrome is partially mediated by the hypofunction of the oxytocinergic system. For clarity, a full list of abbreviations is available in S1 File.

## Materials and methods

### Experimental animals' selection, grouping, and treatment

Twenty-five freshly weaned [25] female Sprague-Dawley rats aged 6–8 weeks, weighing 140-170g, were used as the subjects for the study. Female rats were used

due to their documented increased susceptibility to develop OLZ-induced metabolic syndrome compared to male rats [10]. The rats were fed standard chow pellets (Unga Feeds Kenya Ltd) and water *ad libitum*. The following conditions were maintained in the animal house located in the division of Medical Physiology, Department of Human Anatomy and Medical Physiology, University of Nairobi: Ambient room temperature of (20–25°C). The environment was maintained at a relative humidity of 30–50% with a 12-hour light/dark cycle. The rats were acclimated to the environment and the investigators for one week prior to the commencement of the study's experimental procedures.

The experimental rats were initially assigned into three groups during the induction phase (first 6 weeks): normal control (normal saline) (n=5), low dose OLZ (4 mg/kg) (n=5), and high dose OLZ (8 mg/kg) (n=15). At the end of week six, the normal control and low dose OLZ groups continued into the treatment phase unchanged while the animals in the high dose OLZ group were randomized into the negative control (8 mg/kg of OLZ: n=5), positive control (8 mg/kg OLZ+500 mg/kg of metformin: n=5) and test group (8 mg/kg OLZ and 1 mg/kg OT intraperitoneal injection: n=5); see Fig 1 for details.

## Administration of treatments

OLZ (Square Pharmaceuticals) was administered once daily via oral gavage. The tablets were dissolved in normal saline (0.9% NaCl in water). Two doses were prepared: high dose (8 mg/kg/day) and low dose (4 mg/kg/day). The doses had been predetermined from previous studies [10,11].

A daily injection of 1 mg/kg OT (Wuhan Quanqiling Technology Co., Ltd) was administered intraperitoneally from week 7 to week 12 in the test group. OT powder was dissolved in normal saline. The dose had been predetermined from a previous study carried out at the University of Nairobi, where OT demonstrated significant efficacy in modulating body weight in obese rodents [12].

Metformin hydrochloride (Cosmos Pharmaceuticals, Kenya) 500 milligram tablets were dissolved in normal saline. The rats in the positive control group received a daily dose of 500 mg/kg via oral gavage. This dosage was based on a previous study in which metformin prevented OLZ-induced metabolic dysfunction in rodents [13].

## Parameters assessed

**Random blood glucose and oral glucose tolerance test determination.** Weekly assessments of random blood glucose were conducted using blood samples collected from the lateral tail vein. To minimize pain, topical lidocaine was administered to the tail 15 minutes prior to the procedure. On Call Plus glucometer and glucometer strips were used to measure the blood glucose.

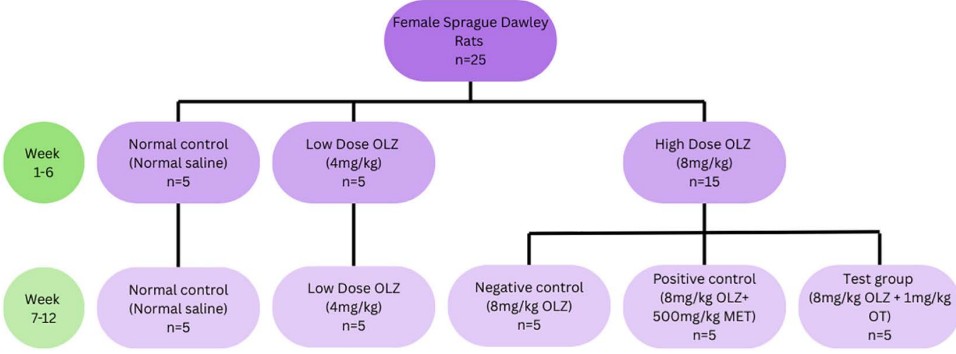

**Fig 1. Schematic overview of the experimental groups.**

Oral glucose tolerance test (for insulin sensitivity assessment) was carried out thrice (on days 42, 63 and 84). The baseline blood glucose was obtained after a 6-hour fast as described for random blood glucose above. The rats then received 2g/kg of glucose by oral gavage. The blood glucose levels at 30-minute intervals after administration of glucose were measured up to the 120th minute [13].

**Lipid profile and adipose tissue weight determination.** The rats were subjected to euthanasia after an overnight fast, administered via an intraperitoneal injection of phenobarbital (6%), at the end of the study. Blood was collected, via cardiac puncture, in a red top vacutainer (no anticoagulant present). The respective collected blood samples were allowed to stand for 45 minutes at room temperature to allow clot formation. The whole blood was centrifuged at 1500 revolutions per minute for 10 minutes to obtain serum. The serum total cholesterol, triglycerides, low-density lipoprotein cholesterol (LDL-C), and high-density lipoprotein cholesterol (HDL-C) were then quantified.

At the end of the twelfth week, the rats were euthanized as described above. Mesenteric, retroperitoneal, and pericardial fat were collected and measured using an analytical balance. The differences in weight between the study groups were obtained.

**Determination of liver weights and hepatic triglycerides.** Upon completion of the study, the rats were subjected to euthanasia following an overnight fast, administered via an intraperitoneal injection of phenobarbital (6%). After making a midline incision, the liver was meticulously extracted from the abdominal cavity, and an analytical balance was used to weigh it. The hepatic index was obtained by dividing the hepatic weight by the rat's body weight (both in grams).

The Butler and Mailing protocol was used to determine the levels of the hepatic triglycerides [14]. Eight (8) milliliters of phosphate buffer were used to homogenize two (2) grams of liver. A one (1) milliliter portion of the homogenate (obtained from the left lateral lobe of each rat) was transferred into a glass cylinder, pre-moistened with two (2) milliliters of chloroform, and mixed with 4 grams of activated charcoal. Eighteen (18) milliliters of chloroform were transferred to the mixture and shaken for approximately ten (10) minutes, then filtered. The filtrate was distributed into three (3) test tubes, with one (1) milliliter of oil being pipetted into each. The test tubes were submerged in an 80°C water bath to facilitate evaporation of chloroform. The first two (2) groups of test tubes were treated with alcoholic potassium hydroxide (0.5 milliliters) while the remaining group was treated with a similar amount of 95% alcohol. The test tubes were left to sit at 70°C for about twenty (20) minutes. After addition of 0.2N sulphuric acid (0.5 milliliters), the test tubes were submerged in a water bath (100°C) for twenty (20) minutes, then left to cool. Thereafter, a succession of chemicals was added at different time frames to each test tube as follows: 0.1 milliliters of periodate solution immediately after cooling, 0.1 milliliters of sodium arsenite after ten (10) minutes, and five (50) milliliters of chromotropic acid after forty-five (45) minutes. They were returned to the water bath for half an hour. The optical densities at 540nm (which corresponds to the absorption maximum of the chromotropic acid-triglyceride complex) were determined. Validation parameters such as linearity, precision, accuracy, and limit of detection were adopted from the Butler and Mailing protocol. The formula used to determine the triglyceride content (in milligrams per gram of tissue) is as follows:

$$\frac{200}{A} \times R \times 0.05 = 10\frac{R}{A}$$

Where R represents:

$$R = \frac{\text{Optical density (O.D)} saponified\ unknown - O.D\ unsaponified\ unknown}{O.D\ saponified\ corn\ oil\ standard - O.D\ unsaponified\ corn\ oil\ standard}$$

A represents: volume of an aliquot of chloroform extract in ml (1 ml was used).

**Liver histopathological examination.** The excised livers were fixed in 10% formalin for two days. Sections were made and deparaffinized by placing them in xylene for fifteen (15) minutes. The sections were then rehydrated by passing

them through a series of descending grades of alcohol and finally to water. The slides were stained with Hematoxylin for one (1) minute, differentiated in 1% acid alcohol, blued, and counterstained with Eosin. The sections were dehydrated using ascending grades of alcohol, cleared using xylene, and dried on filter paper. The levels of steatosis were assessed at ×400 magnification using an Olympus DP71 microscope connected to a video camera. Quantification of score was conducted using Image J™ (National Institutes of Health, Bethesda, MD, USA) analysis software, calculated as steatotic area relative to the total area of each microscopic field and expressed as percentage steatotic area (%). Ten microscopic fields per sample were analyzed.

**Food intake and body weight determination.** Food intake was determined daily per cage for twelve (12) weeks, using a weighing scale. A 500-gram capacity container was placed on a weighing scale and tared to zero. Food pellets were then added to the container and provided to the subjects. After twenty-four (24) hours, any remaining food was collected from the cages, returned to the same container, and weighed. The difference between the initial and final weights was calculated and recorded. Weekly averages were then computed. The body weight of the rats was measured weekly using a weighing scale.

## Ethical consideration

Ethical approval for the experimental study was obtained from the Biosafety, Animal Use and Ethics Committee, Department of Veterinary Anatomy and Physiology, Faculty of Veterinary Medicine, University of Nairobi, and FVM BAUEC/2023/440 was the permit number issued. All surgical procedures were carried out under sodium pentobarbital anesthesia, with rigorous measures taken to reduce distress and discomfort.

## Statistical analysis

All data were expressed as mean±SEM. The data were analyzed using GraphPad Prism 8.0.2 (GraphPad Software, San Diego, CA, USA). Differences among groups were assessed by one-way analysis of variance (ANOVA) followed by Tukey's post hoc multiple comparisons test in cases of significance, which was set at $p < 0.05$.

## Results

### Induction phase outcomes

**Food intake changes during the induction phase.** During the first two weeks of the study, there were no statistically significant differences in food consumption among the three groups.

The differences in food intake among the three experimental groups attained statistical significance, at the end of the third week: [148.1±2.712 grams (normal control group) versus 150.7±2.679 grams (low dose OLZ group) versus 170.7±1.848 grams (high dose OLZ group A) versus 172.3±4.694 grams (high dose OLZ group B) versus 169.6±3.804 grams (high dose OLZ group C): $p < 0.0001$]. Following the primary analysis, Tukey's multiple comparisons test identified significant differences between normal control group and high dose OLZ group A ($p = 0.0003$), normal control group and high dose OLZ group B ($p = 0.0001$), normal control group and high dose OLZ group C ($p = 0.0007$), low dose OLZ group and high dose OLZ group A ($p = 0.0015$), low dose OLZ group and high dose OLZ group B ($p = 0.0006$) and low dose OLZ group and high dose OLZ group C ($p = 0.0029$).

Statistically significant differences in food intake were observed between the three experimental groups, at the end of the fourth week: [150.4±3.637 grams (normal control group) versus 154.1±4.733 grams (low dose OLZ group) versus 177.1±6.273 grams (high dose OLZ group A) versus 175.4±5.200 grams (high dose OLZ group B) versus 176.9±4.877 grams (high dose OLZ group C): $p = 0.0004$]. Following the primary analysis, Tukey's multiple comparisons test identified significant differences between normal control group and high dose OLZ group A ($p = 0.0006$), normal control group and high dose OLZ group B ($p = 0.0112$), normal control group and high dose OLZ group C ($p = 0.0067$), low dose OLZ group

and high dose OLZ group A (p = 0.0225), low dose OLZ group and high dose OLZ group B (p = 0.0399) and low dose OLZ group and high dose OLZ group C (p = 0.0009).

Statistically significant differences in food intake were observed between the three experimental groups, at the end of the fifth week: [149.6 ± 9.010 grams (normal control group) versus 151.7 ± 4.052 grams (low dose OLZ group) versus 179.0 ± 3.331 grams (high dose OLZ group A) versus 180.1 ± 4.044 grams (high dose OLZ group B) versus 177.6 ± 5.529 grams (high dose OLZ group C): p = 0.0002]. Following the primary analysis, Tukey's multiple comparisons test identified significant differences between normal control group and high dose OLZ group A (p = 0.0066), normal control group and high dose OLZ group B (p = 0.0045), normal control group and high dose OLZ group C (p = 0.0105), low dose OLZ group and high dose OLZ group A (p = 0.0132), low dose OLZ group and high dose OLZ group B (p = 0.0092) and low dose OLZ group and high dose OLZ group C (p = 0.0207).

Statistically significant differences in food intake were observed between the three experimental groups, at the end of the sixth week: [148.1 ± 4.818 grams (normal control group) versus 149.9 ± 4.490 grams (low dose OLZ group) versus 178.6 ± 4.571 grams (high dose OLZ group A) versus 179.6 ± 3.772 grams (high dose OLZ group B) versus 180.4 ± 6.082 grams (high dose OLZ group C): p < 0.0001]. Following the primary analysis, Tukey's multiple comparisons test identified significant differences between normal control group and high dose OLZ group A (p = 0.0009), normal control group and high dose OLZ group B (p = 0.0006), normal control group and high dose OLZ group C (p = 0.0004), low dose OLZ group and high dose OLZ group A (p = 0.0018), low dose OLZ group and high dose OLZ group B (p = 0.0012) and low dose OLZ group and high dose OLZ group C (p = 0.0009).

Fig 2 demonstrates the graphical representations of mean food intake at weekly intervals during the induction phase. Food intake data during induction are provided in S2 File.

**Body weight changes during the induction phase.** During the first four weeks of the study, there were no statistically significant differences in body weights among the three groups.

The differences in body weight among the three experimental groups attained statistical significance, at the end of the fifth week: [207.2 ± 4.443 grams (normal control group) versus 207.6 ± 2.502 grams (low dose OLZ group) versus 218.2 ± 2.538 grams (high dose OLZ group A) versus 220.2 ± 3.023 grams (high dose OLZ group B) versus 224.8 ± 2.332 grams (high dose OLZ group C): p = 0.0015]. Following the primary analysis, Tukey's multiple comparisons test identified significant differences between normal control group and high dose OLZ group B (p = 0.0495), normal control group and high dose OLZ group C (p = 0.0049) and low dose OLZ group and high dose OLZ group C (p = 0.0061).

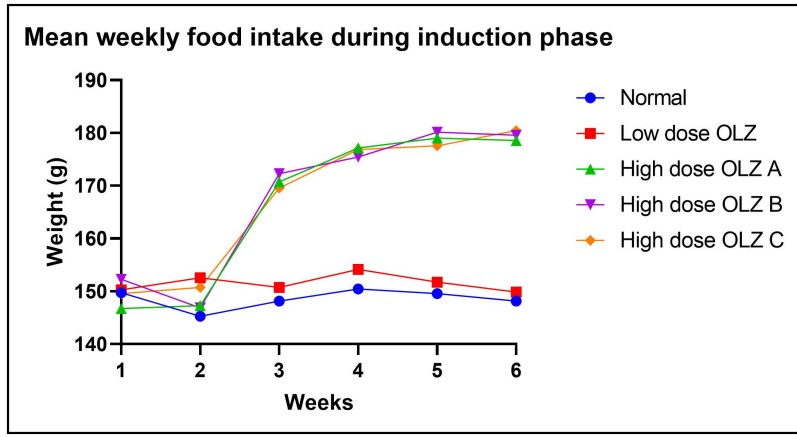

**Fig 2. Line graph illustrating mean food intake (in grams) at weekly intervals during the induction phase.**

Statistically significant differences in body weight were observed between the three experimental groups, at the end of the sixth week: [220.6±2.926 grams (normal control group) versus 221.6±1.631 grams (low dose OLZ group) versus 235.8±4.913 grams (high dose OLZ group A) versus 239.8±2.437 grams (high dose OLZ group B) versus 241.4±1.503 grams (high dose OLZ group C): p<0.0001]. Following the primary analysis, Tukey's multiple comparisons test identified significant differences between normal control group and high dose OLZ group A (p=0.0125), normal control group and high dose OLZ group B (p=0.0015), normal control group and high dose OLZ group C (p=0.0006), low dose OLZ group and high dose OLZ group A (p=0.0211), low dose OLZ group and high dose OLZ group B (p=0.0025) and low dose OLZ group and high dose OLZ group C (p=0.0011). Fig 3 demonstrates the graphical representations of mean body weight at weekly intervals during the induction phase and summary table in S1 Table. Body weight data during induction are provided in S3 File.

**Random blood glucose changes during the induction phase.** During the first four weeks of the study, there were no statistically significant differences in random blood glucose levels among the three groups.

The differences in random blood glucose among the three experimental groups attained statistical significance, at the end of the fifth week: [4.8±0.2 mmol/L (normal control group) versus 5.7±0.2 mmol/L (low dose OLZ group) versus 6.7±0.4 mmol/L (high dose OLZ group A) versus 7.0±0.3 mmol/L (high dose OLZ group B) versus 6.9±0.3 mmol/L (high dose OLZ group C): p<0.0001]. Following the primary analysis, Tukey's multiple comparisons test identified significant differences between normal control group and high dose OLZ group A (p=0.0001), normal control group and high dose OLZ group B (p<0.0001), normal control group and high dose OLZ group C (p<0.0001), low dose OLZ group and high dose OLZ group B (p=0.0135) and low dose OLZ group and high dose OLZ group C (p=0.0260).

Statistically significant differences in random blood glucose were observed between the three experimental groups, at the end of the sixth week: [5.9±0.1 mmol/L (normal control group) versus 5.9±0.2 mmol/L (low dose OLZ group) versus 7.8±0.3 mmol/L (high dose OLZ group A) versus 7.6±0.3 mmol/L (high dose OLZ group B) versus 7.9±0.3 (high dose OLZ group C): p<0.0001]. Following the primary analysis, Tukey's multiple comparisons test identified significant differences between normal control group and high dose OLZ group A (p<0.0001), normal control group and high dose OLZ group B (p=0.0003), normal control group and high dose OLZ group C (p<0.0001), low dose OLZ group and high dose OLZ group A (p=0.0001), low dose OLZ group and high dose OLZ group B (p=0.0004) and low dose OLZ group and

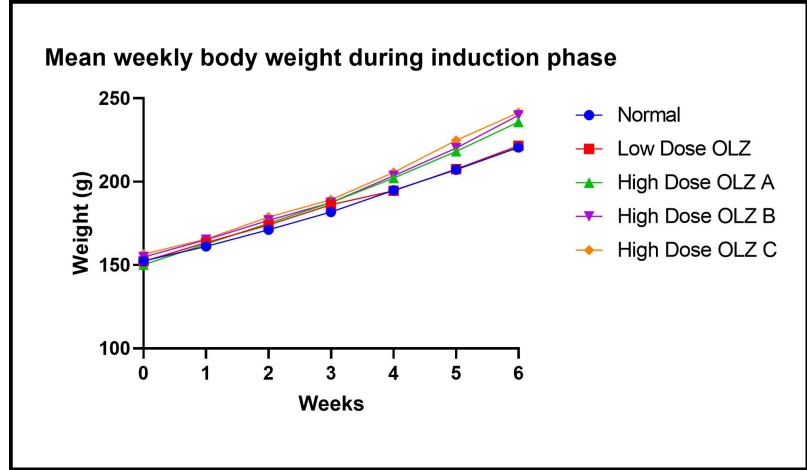

**Fig 3. Line graph illustrating mean body weight (in grams) at weekly intervals during the induction phase.**

high dose OLZ group C (p < 0.0001). Fig 4 demonstrates the graphical representations of mean random blood glucose at weekly intervals during the induction phase. Random blood glucose during induction data are provided in S4 File.

**Oral glucose tolerance test at the end of induction phase (day 42).** Statistically significant differences in the area under the curve values (AUC) were observed between the three experimental groups, at the end of the sixth week: [684.3 ± 27.50 mmol/L.min (normal control group) versus 697.2 ± 12.44 mmol/L.min (low dose OLZ group) versus 850.0 ± 22.53 mmol/L.min (high dose OLZ group A) versus 853.8 ± 24.53 mmol/L.min (high dose OLZ group B) versus 836.7 ± 28.52 mmol/L.min (high dose OLZ group C): p < 0.0001]. Following the primary analysis, Tukey's multiple comparisons test identified significant differences between normal control group and high dose OLZ group A (p = 0.0001), normal control group and high dose OLZ group B (p = 0.0004), normal control group and high dose OLZ group C (p = 0.0004), low dose OLZ group and high dose OLZ group A (p = 0.0004), low dose OLZ group and high dose OLZ group B (p = 0.0015) and low dose OLZ group and high dose OLZ group C (p = 0.0013). Fig 5 demonstrates the graphical representations of mean blood glucose and area under curve at the end of the induction phase (day 42). Oral glucose tolerance test data on day 42 are provided in S5 File.

### Treatment phase outcomes

**Food intake changes during the treatment phase.** Statistically significant differences in food intake were observed between the five experimental groups, at the end of the seventh week: [157.3 ± 4.799 grams (normal control group) versus 158.3 ± 3.130 grams (low dose OLZ group) versus 184.7 ± 4.638 grams (negative control group) versus 176.3 ± 4.341 grams (test group) versus 179.1 ± 4.758 grams (positive control group): p = 0.0001]. Following the primary analysis, Tukey's multiple comparisons test identified significant differences between normal control group and negative control group (p = 0.0010), normal control group and test group (p = 0.0340), normal control group and positive control group (p = 0.0110), low dose OLZ group and negative control group (p = 0.0016), low dose OLZ group and test group (p = 0.0492) and low dose OLZ group and positive control group (p = 0.0165).

Statistically significant differences in food intake were observed between the five experimental groups, at the end of the eigth week: [160.1 ± 3.058 grams (normal control group) versus 159.9 ± 7.236 grams (low dose OLZ group) versus 188.6 ± 5.601 grams (negative control group) versus 170.3 ± 6.611 grams (test group) versus 173.7 ± 3.932 grams (positive control group): p = 0.0053]. Following the primary analysis, Tukey's multiple comparisons test identified significant differences between normal control group and negative control group (p = 0.0083) and low dose OLZ group and negative control group (p = 0.4265).

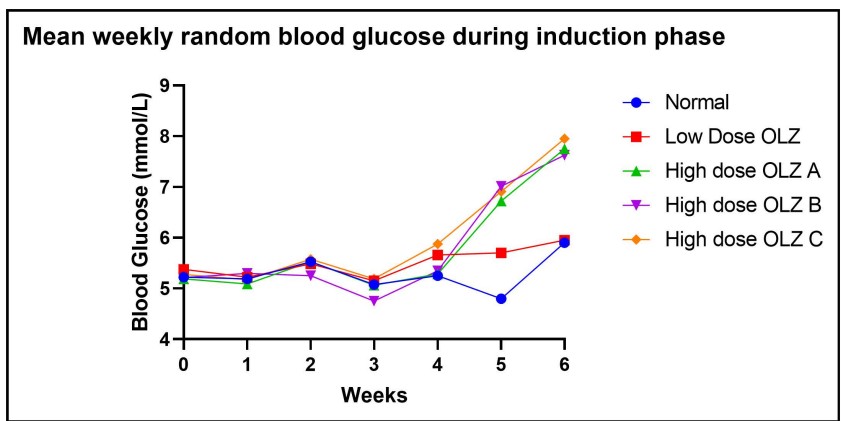

**Fig 4. Line graph illustrating mean random blood glucose (mmol/L) at weekly intervals during the induction phase.**

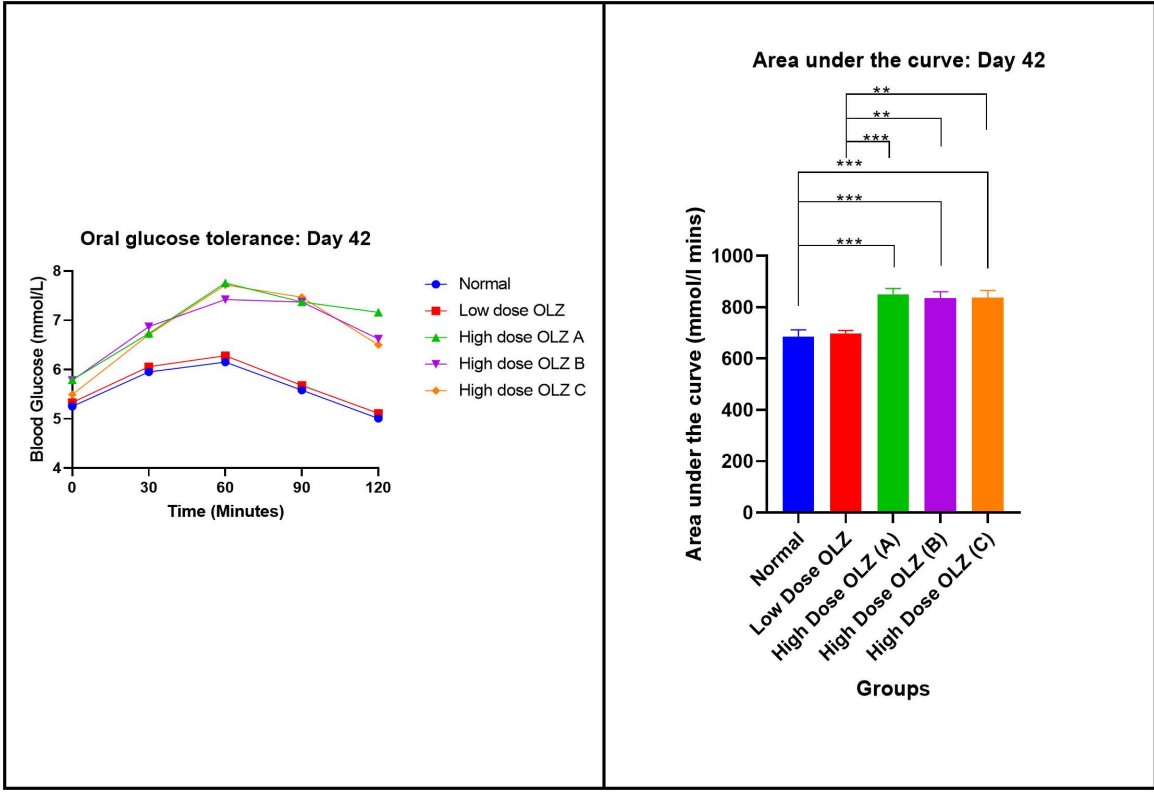

**Fig 5. Mean blood glucose responses (mmol/L) to an oral blood glucose bolus 2g/kg during a 120-minute period and mean area under the curve (mmol/L.min) during the oral blood glucose tolerance test.** Results are expressed as mean±SEM. (**- p<0.01, ***- p<0.001).

Statistically significant differences in food intake were observed between the five experimental groups, at the end of the ninth week: [159.3±5.671 grams (normal control group) versus 163.0±2.960 grams (low dose OLZ group) versus 190.1±1.752 grams (negative control group) versus 161.4±1.913 grams (test group) versus 163.3±3.343 grams (positive control group): p<0.0001]. Following the primary analysis, Tukey's multiple comparisons test identified significant differences between normal control group and negative control group (p<0.0001), low dose OLZ group and negative control group (p<0.0001), test group and negative control group (p<0.0001) and positive control group and negative control group (p<0.0001).

Statistically significant differences in food intake were observed between the five experimental groups, at the end of the tenth week: [160.9±2.721 grams (normal control group) versus 159.9±2.915 grams (low dose OLZ group) versus 196.3±3.421 grams (negative control group) versus 162.4±4.134 grams (test group) versus 160.4±1.131 grams (positive control group): p<0.0001. Following the primary analysis, Tukey's multiple comparisons test identified significant differences between normal control group and negative control group (p<0.0001), low dose OLZ group and negative control group (p<0.0001), test group and negative control group (p<0.0001) and positive control group and negative control group (p<0.0001).

Statistically significant differences in food intake were observed between the five experimental groups, at the end of the eleventh week: [159.9±3.291 grams (normal control group) versus 161.3±2.542 grams (low dose OLZ group) versus 193.7±4.679 grams (negative control group) versus 158.1±2.890 grams (test group) versus 162.7±4.849 grams (positive control group): p<0.0001. Following the primary analysis, Tukey's multiple comparisons test identified significant

differences between normal control group and negative control group (p<0.0001), low dose OLZ group and negative control group (p<0.0001), test group and negative control group (p<0.0001) and positive control group and negative control group (p<0.0001).

Statistically significant differences in food intake were observed between the five experimental groups, at the end of the twelfth week: [160.9±2.040 grams (normal control group) versus 163.7±2.570 grams (low dose OLZ group) versus 196.6±6.125 grams (negative control group) versus 162.6±2.080 grams (test group) versus 164.1±2.632 grams (positive control group): p<0.0001]. Following the primary analysis, Tukey's multiple comparisons test identified significant differences between normal control group and negative control group (p<0.0001), low dose OLZ group and negative control group (p<0.0001), test group and negative control group (p<0.0001) and positive control group and negative control group (p<0.0001). Fig 6 demonstrates the graphical representations of mean food intake at weekly intervals during the treatment phase. Food intake data during treatment are provided in S6 File.

**Body weight changes during the treatment phase.** Statistically significant differences in body weight were observed between the five experimental groups, at the end of the seventh week: [234.6±3.385 grams (normal control group) versus 236.2±3.826 grams (low dose OLZ group) versus 260.6±2.159 grams (negative control group) versus 261.0±1.581 grams (test group) versus 261.6±1.913 grams (positive control group): p<0.0001. Following the primary analysis, Tukey's multiple comparisons test identified significant differences between normal control group and negative control group (p<0.0001), normal control group and test group (p<0.0001), normal control group and positive control group (p<0.0001), low dose OLZ group and negative control group (p<0.0001), low dose OLZ group and test group (p<0.0001) and low dose OLZ group and positive control group (p<0.0001).

Statistically significant differences in body weight were observed between the five experimental groups, at the end of the eigth week: [245.6±1.536 grams (normal control group) versus 248.6±4.308 grams (low dose OLZ group) versus 281.2±2.417 grams (negative control group) versus 274.6±2.088 grams (test group) versus 272.2±4.841 grams (positive control group): p<0.0001]. Following the primary analysis, Tukey's multiple comparisons test identified significant differences between normal control group and negative control group (p<0.0001), normal control group and test group (p<0.0001), normal control group and positive control group (p=0.0001), low dose OLZ group and negative control group (p<0.0001), low dose OLZ group and test group (p=0.0002) and low dose OLZ group and positive control group (p=0.0005).

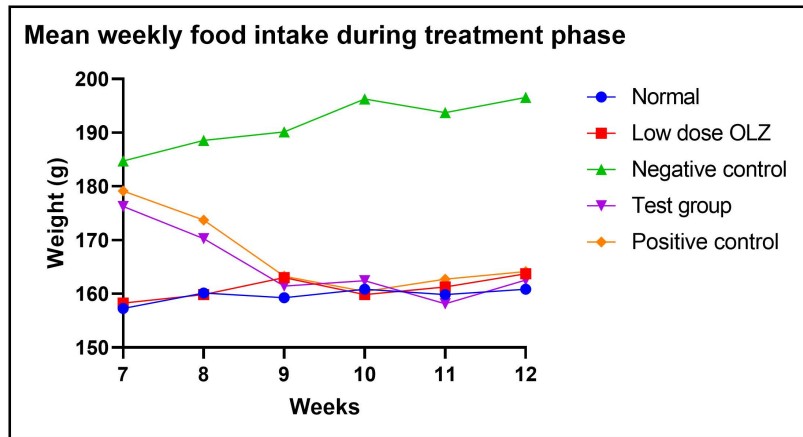

**Fig 6. Line graph illustrating mean food intake (g) at weekly intervals during the treatment phase.**

Statistically significant differences in body weight were observed between the five experimental groups, at the end of the ninth week: [256.2±1.158 grams (normal control group) versus 262.0±3.066 grams (low dose OLZ group) versus 306.0±3.268 grams (negative control group) versus 270.0±2.608 grams (test group) versus 272.2±4.841 grams (positive control group): p<0.0001]. Following the primary analysis, Tukey's multiple comparisons test identified significant differences between normal control group and negative control group (p<0.0001), normal control group and test group (p=0.0218), normal control group and positive control group (p=0.0034), low dose OLZ group and negative control group (p<0.0001), negative control group and test group (p<0.0001) and negative control group and positive control group (p<0.0001).

Statistically significant differences in body weight were observed between the five experimental groups, at the end of the tenth week: [268.8±0.8602 grams (normal control group) versus 274.8±3.693 grams (low dose OLZ group) versus 336.0±4.037 grams (negative control group) versus 274.8±2.746 grams (test group) versus 275.4±1.435 grams (positive control group): p<0.0001]. Following the primary analysis, Tukey's multiple comparisons test identified significant differences between normal control group and negative control group (p<0.0001), low dose OLZ group and negative control group (p<0.0001), test group and negative control group (p<0.0001) and positive control group and negative control group (p<0.0001).

Statistically significant differences in body weight were observed between the five experimental groups, at the end of the eleventh week: [280.6±2.857 grams (normal control group) versus 288.4±1.077 grams (low dose OLZ group) versus 366.8±4.409 grams (negative control group) versus 287.6±1.327 grams (test group) versus 288.4±2.315 grams (positive control group): p<0.0001]. Following the primary analysis, Tukey's multiple comparisons test identified significant differences between normal control group and negative control group (p<0.0001), low dose OLZ group and negative control group (p<0.0001), test group and negative control group (p<0.0001) and positive control group and negative control group (p<0.0001).

Statistically significant differences in body weight were observed between the five experimental groups, at the end of the twelfth week: [287.4±2.159 grams (normal control group) versus 297.2±1.463 grams (low dose OLZ group) versus 376.2±5.919 grams (negative control group) versus 297.2±1.960 grams (test group) versus 299.4±6.794 grams (positive control group): p<0.0001]. Following the primary analysis, Tukey's multiple comparisons test identified significant differences between normal control group and negative control group (p<0.0001), low dose OLZ group and negative control group (p<0.0001), test group and negative control group (p<0.0001) and positive control group and negative control group (p<0.0001). Fig 7 demonstrates the graphical representations of mean body weight at weekly intervals during the treatment phase and summary table in S2 Table. Body weight data during treatment are provided in S7 File.

**Random blood glucose changes during the treatment phase.** Statistically significant differences in random blood glucose were observed between the five experimental groups, at the end of the seventh week: [5.2±0.2 mmol/L (normal control group) versus 5.3±0.2 mmol/L (low dose OLZ group) versus 7.8±0.2 mmol/L (negative control group) versus 7.7±0.2 mmol/L (test group) versus 7.7±0.2 mmol/L (positive control group): p<0.0001]. Following the primary analysis, Tukey's multiple comparisons test identified significant differences between normal control group and negative control group (p<0.0001), normal control group and test group (p<0.0001), normal control group and positive control group (p<0.0001), low dose OLZ group and negative control (p<0.0001), low dose OLZ group and test group (p<0.0001) and low dose OLZ group and positive control (p<0.0001).

Statistically significant differences in random blood glucose were observed between the five experimental groups, at the end of the eigth week: [5.5±0.3 mmol/L (normal control group) versus 5.7±0.2 mmol/L (low dose OLZ group) versus 7.9±0.2 mmol/L (negative control group) versus 6.5±0.2 mmol/L (test group) versus 6.6±0.3 mmol/L (positive control group): p<0.0001]. Following the primary analysis, Tukey's multiple comparisons test identified significant differences between normal control group and negative control group (p<0.0001), normal control group and test group (p=0.0343), normal control group and positive control group (p=0.0102), low dose OLZ group and negative control (p<0.0001), low

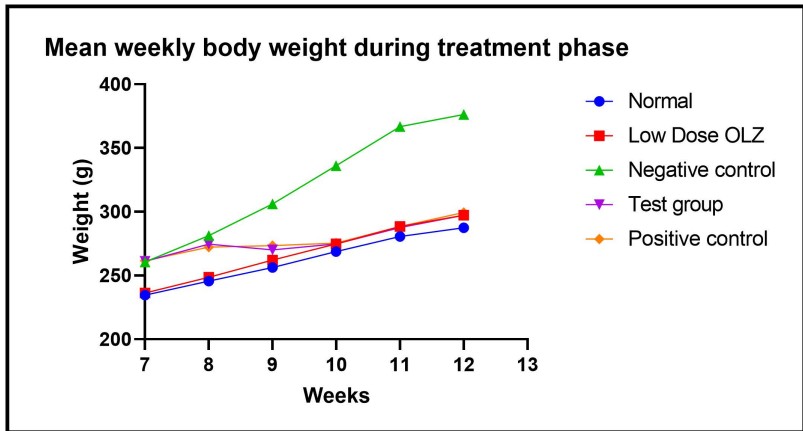

**Fig 7. Line graph illustrating mean body weight (in grams) at weekly intervals during the treatment phase.**

dose OLZ group and positive ($p < 0.0001$), negative control and test group ($p = 0.0003$) and, negative control and positive control ($p = 0.0013$).

Statistically significant differences in random blood glucose were observed between the five experimental groups, at the end of the ninth week: [$5.4 \pm 0.2$ mmol/L (normal control group) versus $5.6 \pm 0.2$ mmol/L (low dose OLZ group) versus $8.7 \pm 0.2$ mmol/L (negative control group) versus $5.6 \pm 0.3$ mmol/L (test group) versus $5.6 \pm 0.3$ mmol/L (positive control group): $p < 0.0001$]. Following the primary analysis, Tukey's multiple comparisons test identified significant differences between normal control group and negative control group ($p < 0.0001$), low dose OLZ group and negative control group ($p < 0.0001$), negative control and test group ($p < 0.0001$) and, negative control and positive control ($p < 0.0001$).

Statistically significant differences in random blood glucose were observed between the five experimental groups, at the end of the tenth week: [$5.6 \pm 0.2$ mmol/L (normal control group) versus $5.8 \pm 0.3$ mmol/L (low dose OLZ group) versus $9.2 \pm 0.4$ mmol/L (negative control group) versus $5.8 \pm 0.2$ mmol/L (test group) versus $5.7 \pm 0.2$ mmol/L (positive control group): $p < 0.0001$]. Following the primary analysis, Tukey's multiple comparisons test identified significant differences between normal control group and negative control group ($p < 0.0001$), low dose OLZ group and negative control ($p < 0.0001$), negative control and test group ($p < 0.0001$) and, negative control and positive control ($p < 0.0001$).

Statistically significant differences in random blood glucose were observed between the five experimental groups, at the end of the eleventh week: [$5.7 \pm 0.3$ mmol/L (normal control group) versus $5.8 \pm 0.3$ mmol/L (low dose OLZ group) versus $9.6 \pm 0.3$ mmol/L (negative control group) versus $5.8 \pm 0.2$ mmol/L (test group) versus $5.9 \pm 0.2$ mmol/L (positive control group): $p < 0.0001$]. Following the primary analysis, Tukey's multiple comparisons test identified significant differences between normal control group and negative control group ($p < 0.0001$), low dose OLZ group and negative control ($p < 0.0001$), negative control and test group ($p < 0.0001$) and, negative control and positive control ($p < 0.0001$).

Statistically significant differences in random blood glucose were observed between the five experimental groups, at the end of the twelfth week: [$5.5 \pm 0.2$ mmol/L (normal control group) versus $5.7 \pm 0.3$ mmol/L (low dose OLZ group) versus $9.8 \pm 0.2$ mmol/L (negative control group) versus $5.7 \pm 0.3$ mmol/L (test group) versus $5.6 \pm 0.32$ mmol/L (positive control group): $p < 0.0001$]. Following the primary analysis, Tukey's multiple comparisons test identified significant differences between normal control group and negative control group ($p < 0.0001$), low dose OLZ group and negative control ($p < 0.0001$), negative control and test group ($p < 0.0001$) and, negative control and positive control ($p < 0.0001$). Fig 8 demonstrates the graphical representations of mean random blood glucose at weekly intervals during the treatment phase. Random blood glucose data during treatment are provided in S8 File.

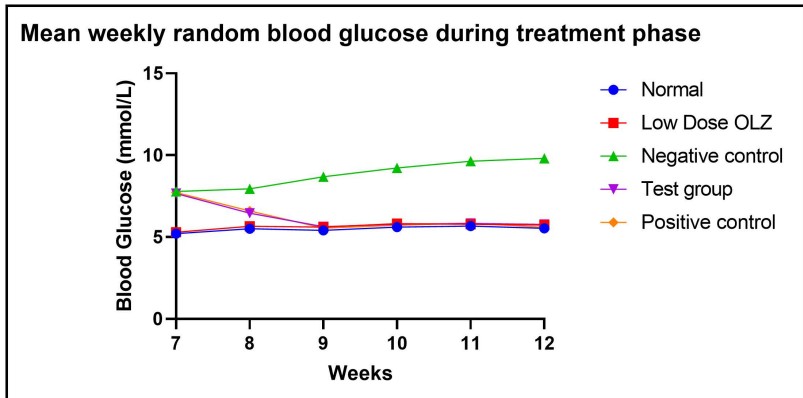

**Fig 8. Line graph illustrating mean random blood glucose (mmol/L) at weekly intervals during the treatment phase.**

**Oral glucose tolerance test during the treatment phase (day 63).** Statistically significant differences in the area under the curve values (AUC) were observed between the five experimental groups, at the end of the ninth week: [683.8±26.74 mmol/L.min (normal control group) versus 696.9±23.53 mmol/L.min (low dose OLZ group) versus 848.8±9.038 mmol/L.min (negative control group) versus 822.2±18.33 mmol/L.min (test group) versus 823.7±17.77 mmol/L.min (positive control group): p<0.0001]. Following the primary analysis, Tukey's multiple comparisons test identified significant differences between normal control group and negative control group (p<0.0001), normal control group and test group (p=0.0002), normal control group and positive control group (p=0.0002), low dose OLZ group and negative control group (p<0.0001), low dose OLZ group and test group (p=0.0008) and low dose OLZ group and positive control group (p=0.0007). Oral glucose tolerance test data on day 63 are provided in S9 File.

**Oral glucose tolerance test at the end of treatment phase (day 84).** Statistically significant differences in the area under the curve values (AUC) were observed between the five experimental groups, at the end of the twelfth week: [685.3±14.74 mmol/L.min (normal control group) versus 698.8±19.18 mmol/L.min (low dose OLZ group) versus 853.3±9.339 mmol/L.min (negative control group) versus 711.6±14.24 mmol/L.min (test group) versus 704.1±16.27 mmol/L.min (positive control group): p<0.0001 Following the primary analysis, Tukey's multiple comparisons test identified significant differences between normal control group and negative control group (p<0.0001), low dose OLZ group and negative control group (p<0.0001), test group and negative control group (p<0.0001) and positive control group and negative control group (p<0.0001). Fig 9 demonstrates the graphical representations of mean blood glucose and area under curve mid-treatment (day 63) and at the end of treatment phase (day 84). Oral glucose tolerance test data on day 84 are provided in S10 File.

**Liver weight, hepatic index, hepatic triglycerides and steatosis at the end of the treatment phase.** Statistically significant differences in the liver weight were observed between the five experimental groups, at the end of the twelfth week: [6.042±0.4280 grams (normal control group) versus 6.816±0.3075 grams (low dose OLZ group) versus 11.16±0.3236 grams (negative control group) versus 6.652±0.2040 grams (test group) versus 6.770±0.4701 grams (positive control group): p<0.0001]. Following the primary analysis, Tukey's multiple comparisons test identified significant differences between normal control group and negative control group (p<0.0001), low dose OLZ group and negative control group (p<0.0001), test group and negative control group (p<0.0001) and positive control group and negative control group (p<0.0001). Liver weight data are provided in S11 File.

Statistically significant differences in the liver to body weight ratio were observed between the five experimental groups, at the end of the twelfth week: [0.02105±0.001605 (normal control group) versus 0.02292±0.0009424 (low

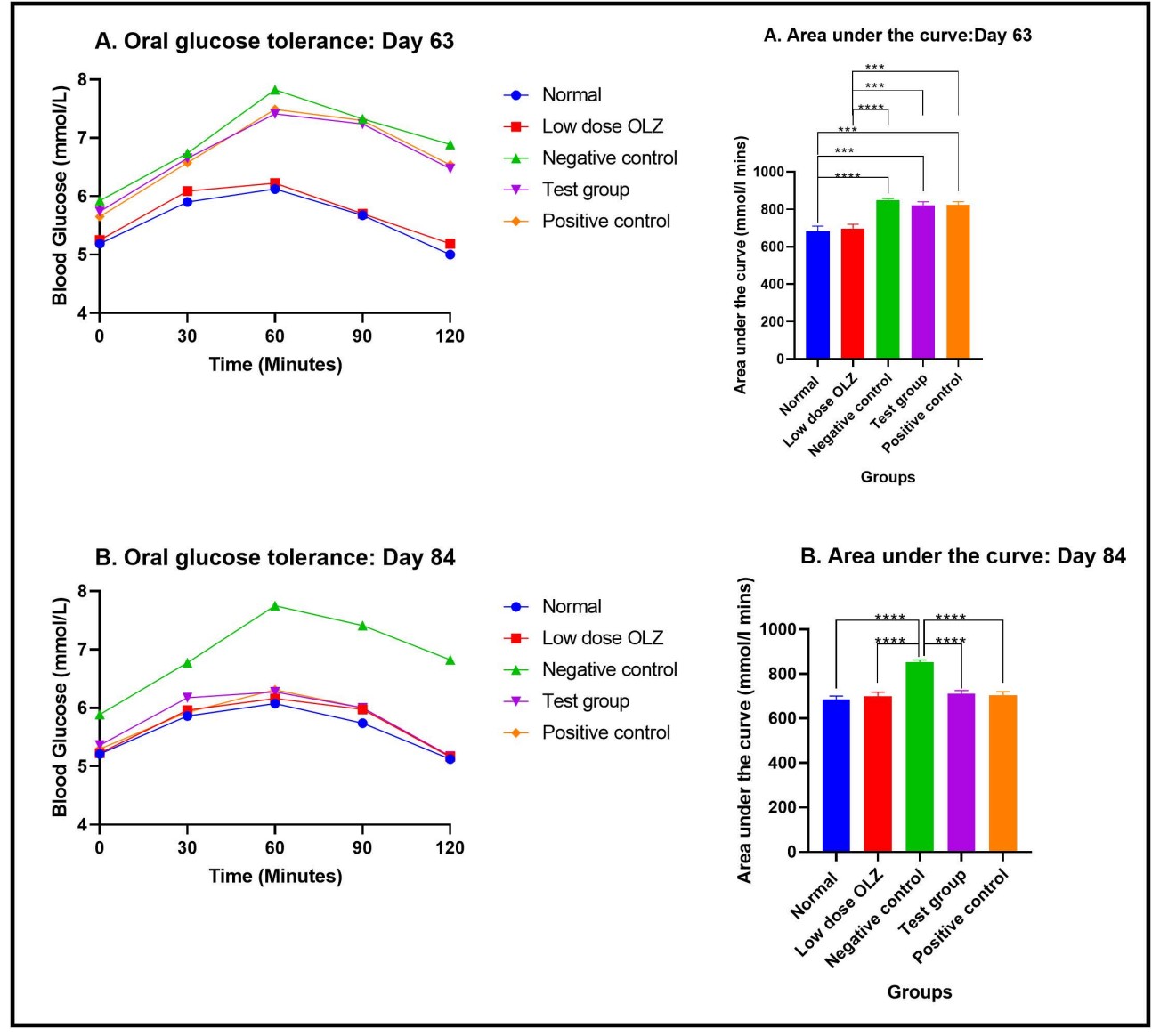

**Fig 9. Mean blood glucose responses (mmol/L) to an oral blood glucose bolus 2g/kg during a 120-minute period and mean area under the curve (mmol/L.min) during the oral blood glucose tolerance test.** Results are expressed as mean±SEM. (***- p<0.001, ****- p<0.0001). A (day 63), B (day 84).

dose OLZ group) versus 0.02971±0. 001021 (negative control group) versus 0.02240±0.0008007 (test group) versus 0.02274±0.001937 (positive control group): p=0.0016]. Following the primary analysis, Tukey's multiple comparisons test identified significant differences between normal control group and negative control group (p=0.0015), low dose OLZ group and negative control group (p=0.0137), test group and negative control group (p=0.0075) and positive control group and negative control group (p=0.0111). Hepatic index data are provided in S12 File.

Statistically significant differences in the hepatic triglycerides content were observed between the five experimental groups, at the end of the twelfth week: [4.240±0.3134 mg/g (normal control group) versus 4.300±0.3799 mg/g (low dose OLZ group) versus 7.910±0.4399 mg/g (negative control group) versus 4.190±0.2610 mg/g (test group) versus

4.250±0.1984 mg/g (positive control group): p<0.0001]. Following the primary analysis, Tukey's multiple comparisons test identified significant differences between normal control group and negative control group (p<0.0001), low dose OLZ group and negative control group (p<0.0001), test group and negative control group (p<0.0001) and positive control group and negative control group (p<0.0001). Hepatic triglycerides data are provided in S13 File.

Statistically significant differences in the hepatic steatosis were observed between the five experimental groups, at the end of the twelfth week: [2.580±0.6902% (normal control group) versus 4.160±0.4366% (low dose OLZ group) versus 24.90±3.355% (negative control group) versus 9.680±1.161% (test group) versus 9.420±1.199% (positive control group): p<0.0001]. Following the primary analysis, Tukey's multiple comparisons test identified significant differences between normal control group and negative control group (p<0.0001), low dose OLZ group and negative control group (p<0.0001), test group and negative control group (p<0.0001) and positive control group and negative control group (p<0.0001). Fig 10 demonstrates the graphical representations of mean liver weights, liver weight to body weight ratio, hepatic triglycerides and hepatic steatosis quantification upon completion of treatment phase. Representative liver histology of the normal control, low dose OLZ, negative control, test, and positive control groups is shown in Fig 11. Hepatic steatosis data are provided in S14 File.

**Visceral adipose tissue at the end of the treatment phase.** Statistically significant differences in the mesenteric adipose tissue weight were observed between the five experimental groups, at the end of the twelfth week: [1.936±0.02315 grams (normal control group) versus 2.452±0.1998 grams (low dose OLZ group) versus 3.450±0.4056 grams (negative control group) versus 1.640±0.1973 grams (test group) versus 1.944±0.1839 grams (positive control group): p=0.0002]. Following the primary analysis, Tukey's multiple comparisons test identified significant differences between normal control group and negative control group (p=0.0017), low dose OLZ group and negative control group (p=0.0498), test group and negative control group (p=0.0002) and positive control group and negative control group (p=0.0018). Mesenteric adipose tissue weight data are provided in S15 File.

Statistically significant differences in the retroperitoneal adipose tissue weight were observed between the five experimental groups, at the end of the twelfth week: [1.664±0.1925 grams (normal control group) versus 1.890±0.1856 grams (low dose OLZ group) versus 3.332±0.6291 grams (negative control group) versus 1.466±0.2332 grams (test group) versus 1.504±0.1461 grams (positive control group): p=0.0035]. Following the primary analysis, Tukey's multiple comparisons test identified significant differences between normal control group and negative control group (p=0.0144), low dose OLZ group and negative control group (p=0.0405), test group and negative control group (p=0.0056) and positive control group and negative control group (p=0.0067). Retroperitoneal adipose tissue weight data are provided in S16 File.

Statistically significant differences in the pericardial adipose tissue weight were observed between the five experimental groups, at the end of the twelfth week: [0.08600±0.02421 grams (normal control group) versus 0.09800±0.03308 grams (low dose OLZ group) versus 0.2200±0.03742 grams (negative control group) versus 0.08800±0.02354 grams (test group) versus 0.09200±0.01200 grams (positive control group): p=0.0100]. Following the primary analysis, Tukey's multiple comparisons test identified significant differences between normal control group and negative control group (p=0.0191), low dose OLZ group and negative control group (p=0.0370), test group and negative control group (p=0.0214) and positive control group and negative control group (p=0.0267). Pericardial adipose tissue weight data are provided in S17 File.

Statistically significant differences in the total adipose tissue weight were observed between the five experimental groups, at the end of the twelfth week: [3.686±0.1894 grams (normal control group) versus 4.440±0.3244 grams (low dose OLZ group) versus 7.002±1.056 grams (negative control group) versus 3.158±0.3905 grams (test group) versus 3.540±0.2554 grams (positive control group): p=0.0005]. Following the primary analysis, Tukey's multiple comparisons test identified significant differences between normal control group and negative control group (p=0.0028), low dose OLZ group and negative control group (p=0.0243), test group and negative control group (p=0.0006) and positive control group and negative control group (p=0.0018). Fig 12 demonstrates the graphical representations of adipose tissue weights upon completion of treatment phase. Total visceral adipose tissue weight data are provided in S18 File.

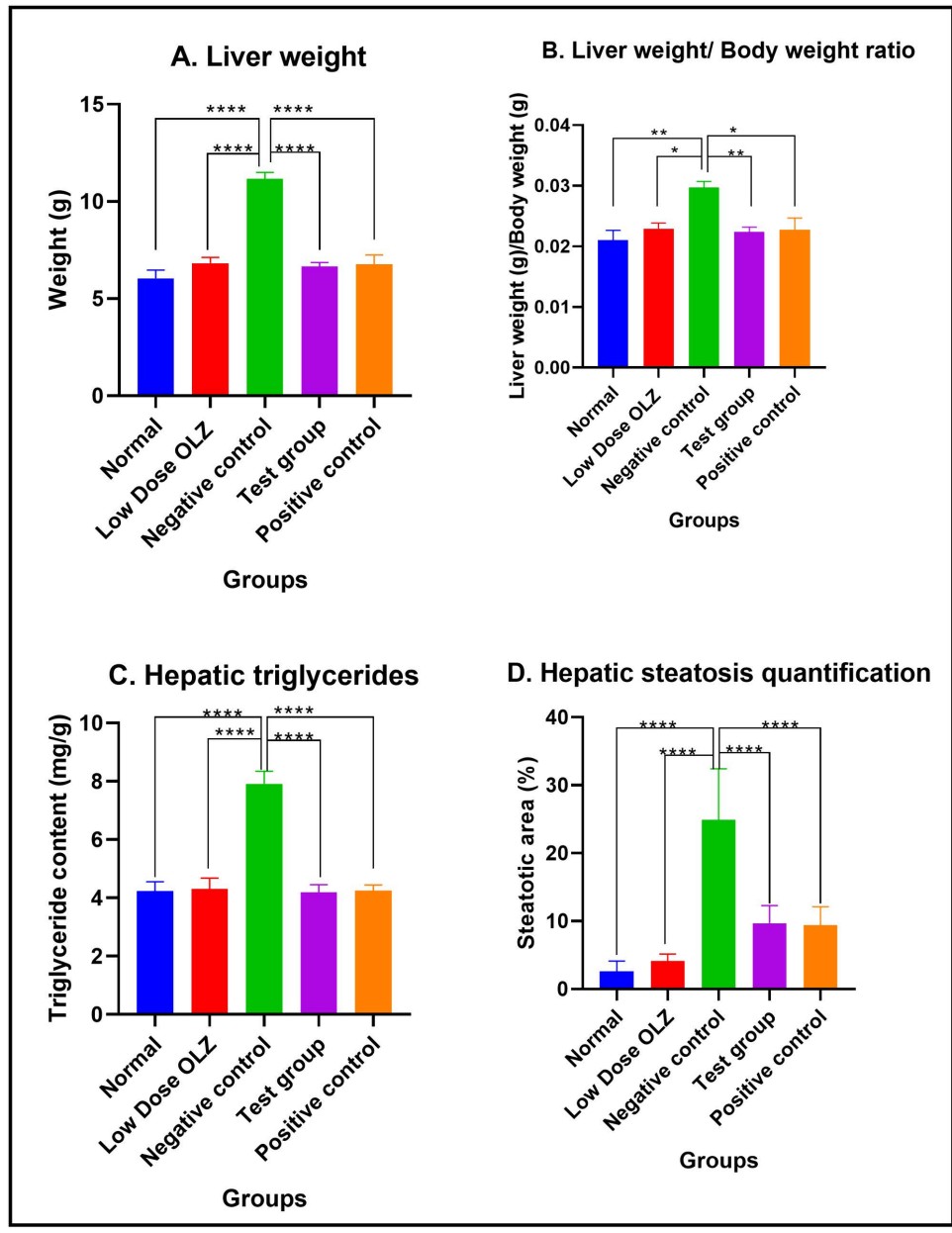

**Fig 10. Mean liver weight (g) (A), liver weight/body ratio (B), hepatic triglyceride (mg/g) (C) and hepatic steatosis quantification (% area) (D) upon completion of the treatment phase.** Results are expressed as mean±SEM. (*- p<0.05, **- p<0.01, ****- p<0.0001).

**Lipid profile at the end of the treatment phase.** Statistically significant differences in the total serum cholesterol were observed between the five experimental groups, at the end of the twelfth week: [5.180±0.1655 mmol/L (normal control group) versus 5.620±0.2396 mmol/L (low dose group) versus 7.240±0.1077 mmol/L (negative control group) versus 5.640±0.3010 mmol/L (test group) versus 5.760±0.3628 mmol/L (positive control group): p=0.0002]. Following the primary analysis, Tukey's multiple comparisons test identified significant differences between normal control group and negative control group (p=0.0001), low dose group and negative control group (p=0.0017), test group and negative

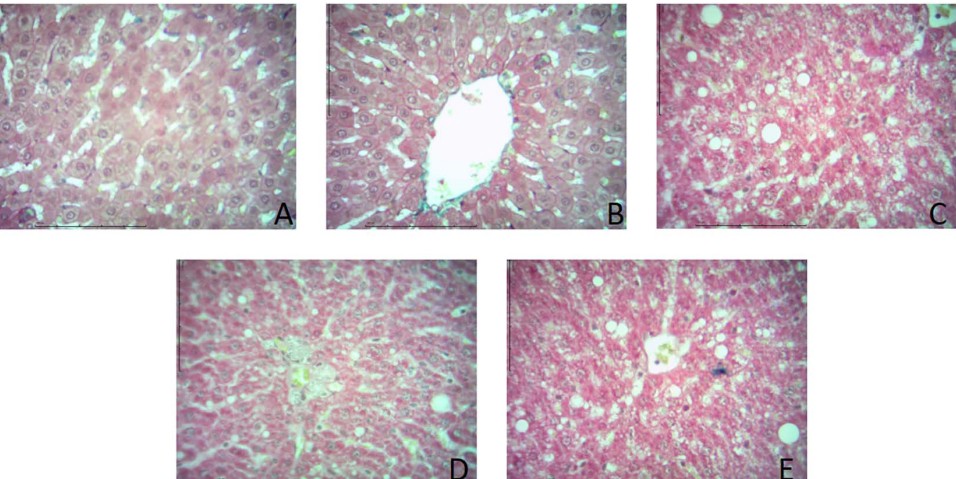

**Fig 11. Representative liver histology of normal control (A), low-dose Olanzapine (B), negative control (C), test (D), and positive control (E) groups (Hematoxylin and Eosin stain, ×400 magnification, scale bar = 100 μm).** Occasional clear vacuoles are present in all groups, with a marked increase in the negative control group. Quantitative analysis using ImageJ (% steatotic area) more accurately reflects differences in hepatic fat accumulation as demonstrated in Fig 10D.

control group (p = 0.0019) and positive control group and negative control group (p = 0.0041). Total serum cholesterol data are provided in S19 File.

Statistically significant differences in the serum triglycerides were observed between the five experimental groups, at the end of the twelfth week: [2.800 ± 0.3661 mmol/L (normal control group) versus 3.040 ± 0.1778 mmol/L (low dose group) versus 6.780 ± 0.2417 mmol/L (negative control group) versus 3.100 ± 0.2214 mmol/L (test group) versus 3.080 ± 0.1393 mmol/L (positive control group): p < 0.0001]. Following the primary analysis, Tukey's multiple comparisons test identified significant differences between normal control group and negative control group (p < 0.0001), low dose group and negative control group (p < 0.0001), test group and negative control group (p < 0.0001) and positive control group and negative control group (p < 0.0001). Serum triglyceride levels data are provided in S20 File.

Statistically significant differences in LDL-C were observed between the five experimental groups, at the end of the twelfth week: [1.920 ± 0.09695 mmol/L (normal control group) versus 2.100 ± 0.1414 mmol/L (low dose group) versus 5.840 ± 0.2960 mmol/L (negative control group) versus 2.320 ± 0.1715 mmol/L (test group) versus 2.280 ± 0.3098 mmol/L (positive control group): p < 0.0001]. Following the primary analysis, Tukey's multiple comparisons test identified significant differences between normal control group and negative control group (p < 0.0001), low dose group and negative control group (p < 0.0001), test group and negative control group (p < 0.0001) and positive control group and negative control group (p < 0.0001). LDL-C data are provided in S21 File.

Statistically significant differences in HDL-C were observed between the five experimental groups, at the end of the twelfth week: [2.020 ± 0.2518 mmol/L (normal control group) versus 1.800 ± 0.1517 mmol/L (low dose group) versus 0.8200 ± 0.1319 mmol/L (negative control group) versus 1.820 ± 0.1828 mmol/L (test group) versus 1.820 ± 0.05831 mmol/L (positive control group): p = 0.0005]. Following the primary analysis, Tukey's multiple comparisons test identified significant differences between normal control group and negative control group (p = 0.0005), low dose group and negative control group (p = 0.0042), test group and negative control group (p = 0.0035) and positive control group and negative control group (p = 0.0035). Fig 13 demonstrates the graphical representations of lipid profile upon completion of treatment phase. HDL-C data are provided in S22 File. Lipid profile summary table in S3 Table.

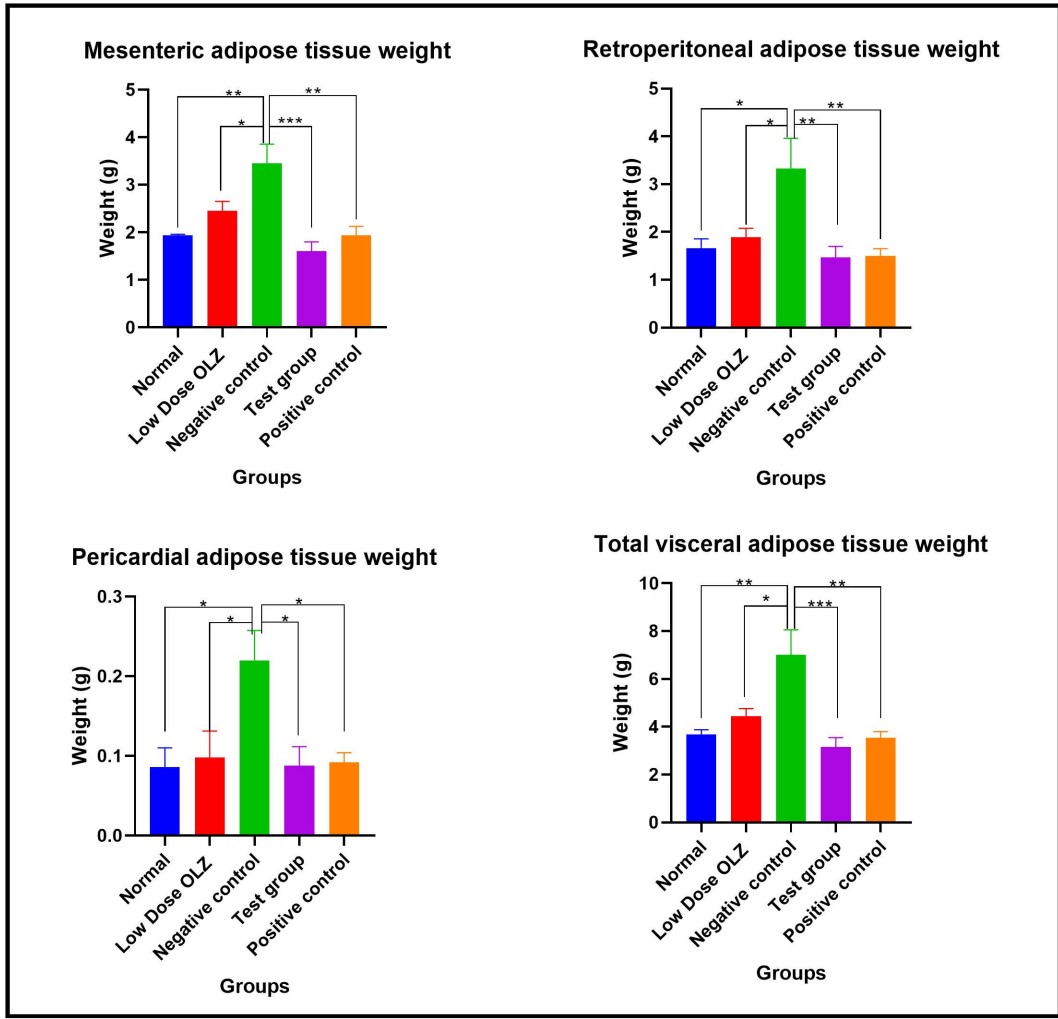

**Fig 12. Mean mesenteric, retroperitoneal, pericardial, and total visceral adipose tissue weights (in grams) upon completion of the experiment.** Results are expressed as mean±SEM. (*- p<0.05, **- p<0.01, ***- p<0.001).

## Discussion

Olanzapine (OLZ) is an atypical antipsychotic that has been connected to the emergence of metabolic abnormalities [5]. Oxytocin (OT), an endogenous hormone, has been suggested to have a role in energy metabolism and reduction in feeding behavior [6]. This study aimed to assess whether oxytocinergic system hypofunction may be partially involved in the pathophysiology of OLZ-induced metabolic dysfunction.

OT significantly attenuated the dose-dependent orexigenic effects of OLZ. These orexigenic effects have been previously reported to be primarily driven by elevated appetite and food intake, rather than decreased activity or expenditure levels, both in healthy and schizophrenic patients [15,16]. The orexigenic effects of OLZ have been linked to several pathways, including interactions with histaminergic [17], serotonergic [18,19], ghrelin [20], and cannabinoid systems [21]. However, the most consistently observed pathway involves the histamine H1 receptor (H1R) and ghrelin receptor (GHSR1a). Disruption of the interaction between H1R and GHSR1a in hypothalamic neurons, with downstream activation of adenosine monophosphate-activated protein kinase (AMPK) and neuropeptide Y (NPY)/ agouti-related peptide (AgRP) signaling,

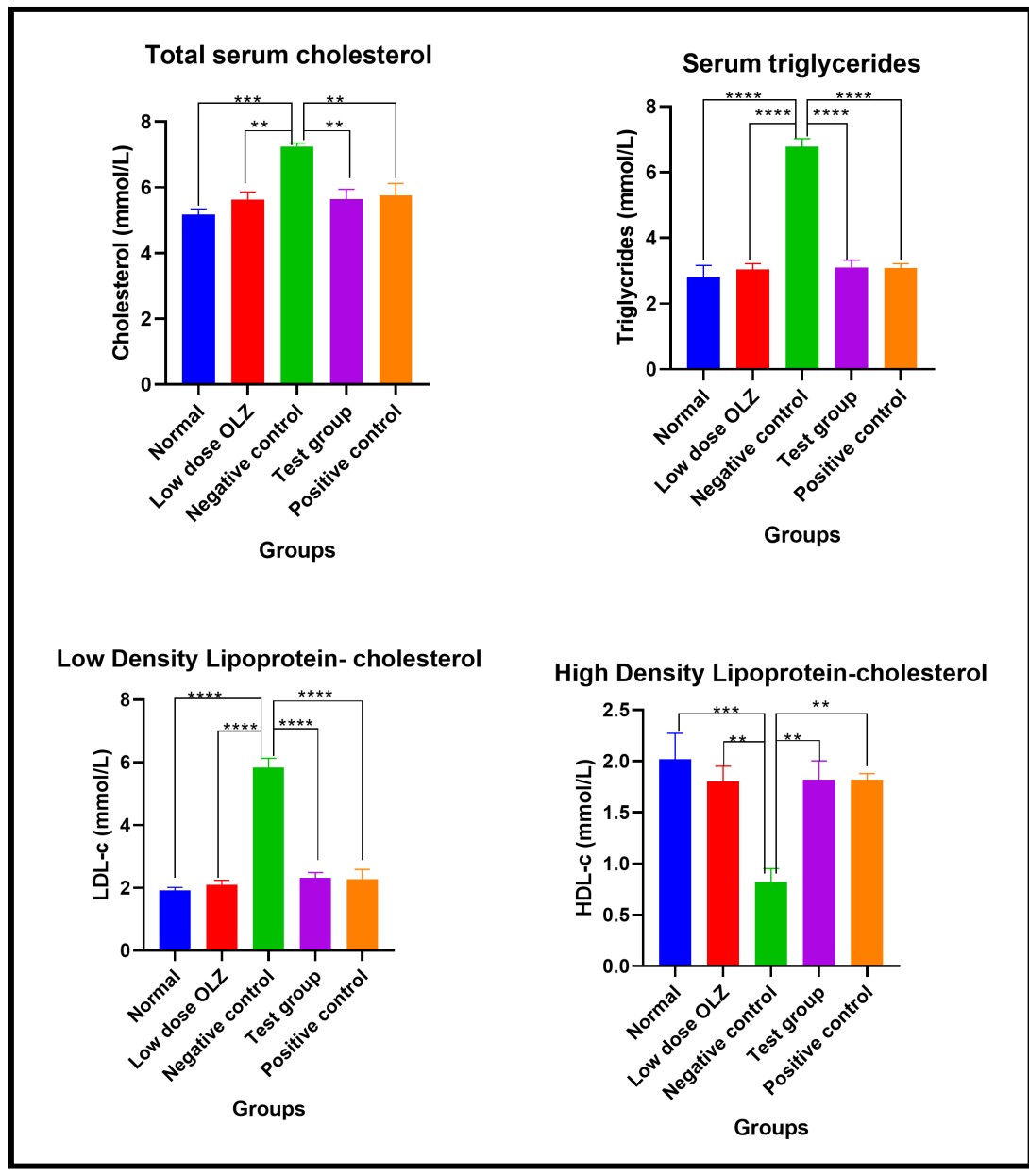

**Fig 13. Mean lipid profile (mmol/L) upon completion of the experimental period.** Results are expressed as mean±SEM. (**- p<0.01, ***- p<0.001, ****- p<0.0001).

has been associated with olanzapine-induced hyperphagia [22]. Further, OLZ has been found to activate hypothalamic AMPK and elevate NPY expression via H1R and 5-HT2C receptor antagonism [23]. Emerging research further supports the central role of AgRP neurons in integrating metabolic cues to maintain energy balance [24]. Consequently, activation of NPY/AgRP neurons is recognized as a key driver of feeding behavior, strongly promoting food intake [25].

In contrast, central OT has been shown to exert anorexigenic effects through direct oxytocinergic pathways originating from the paraventricular and supraoptic nuclei. These pathways project to OXTR-expressing pro-opiomelanocortin

(POMC) neurons in the arcuate nucleus, mediating satiety [26]. OT pathways also reach the ventral tegmental area and nucleus accumbens, modulating reward-driven feeding [27]. They project to the central amygdala, influencing the pleasurable value of food [28,29], and the nucleus tractus solitarius, integrating gut-derived signals from vagal afferents, hormones, and nutrients such as glucose to regulate energy balance [30,31]. The foregoing discussion indicates that OT mediated its anorexigenic effects both at hypothalamic as well as extrahypothalamic sites.

OT significantly attenuated the dose-dependent weight gain effects of OLZ. Part of this weight gain may be explained by increased food intake [32], consistent with the orexigenic effects discussed above, in addition to upregulation of inflammation via elevated pro-inflammatory markers and activation of the IKKβ/NF-κB signaling pathway [33]. Conversely, OT has been shown to induce weight loss by counteracting leptin in diet-induced obese rodents and non-human primates by reducing food intake and boosting energy expenditure [34,35].

OT significantly attenuated the dose-dependent effects of OLZ on central adiposity (i.e., significant increases in mesenteric, retroperitoneal, pericardial, and total visceral fat mass). OLZ has previously been shown to cause increased abdominal fat mass, which was also accompanied by adipocyte enlargement in rodents as well as in humans [36,37]. OLZ has been found to promote adiposity by enhancing lipogenesis via Sterol Regulatory Element-Binding Protein 1 (SREBP-1) activation, leading to the overexpression of fatty acid synthase, resulting in triglyceride accumulation and ultimately adipocyte hypertrophy [5]. In addition, OLZ was also shown to inhibit lipolysis, which accentuates its effects on the stimulation of lipogenesis [38]. In contrast, OT promotes lipolysis and β-oxidation of fatty acids, resulting in reduced adiposity [39]. It is noteworthy that these effects on adipose tissue are independent of its effects on food intake [35,40,41].

OT significantly attenuated the dose-dependent OLZ-induced increases in liver weight, hepatic index, steatosis, and triglyceride levels. OLZ has been shown to upregulate lipid anabolism in hepatocytes [42], leading to steatosis and grossly, increased liver weight [43]. The proposed mechanism via which OLZ induces hepatocyte lipid accumulation is via inhibition of hepatic AMPK, which normally inhibits mTORC1 (mechanistic target of rapamycin complex 1)-SREBP1c-mediated lipogenesis [5,44]. OLZ also downregulates peroxisome proliferator-activated receptor (PPAR), thereby reducing lipolysis and β-oxidation, leading to hepatic steatosis [45]. OT treatment, on the other hand, has been reported to ameliorate hepatic steatosis and reduce triglyceride accumulation, potentially through hepatocyte OXTR signaling, which promotes autophagy and hepatocyte regeneration [46,47].

OT significantly attenuated OLZ's deleterious effects on glycemic control in this study. OLZ has previously been shown to reduce glucose uptake and increase plasma insulin levels compared with baseline levels [48]. These effects may be reversible after discontinuation of OLZ [49]. OLZ has been associated with postprandial elevations in insulin, glucagon-like peptide 1 (GLP-1), and glucagon, consistent with insulin resistance [50]. Mechanistically, OLZ has been found to impair glycogen synthesis in hepatocytes and skeletal muscle by disrupting insulin signaling, including reduced phosphorylation of insulin receptor substrate-1 and glycogen synthase kinase-3 (GSK-3α/β), and attenuating AKT [51]. Furthermore, interference with serotonin signaling may impair GLUT4 translocation through disruption of Rab4 serotonylation, further contributing to dysglycemia [48].

In contrast, OT has well-documented effects on improving glycemic control. It stimulates insulin secretion and enhances insulin sensitivity in both animal models and humans [6]. OT also promotes glucose uptake in skeletal muscle, and chronic administration improves glucose intolerance, fatty liver, cardiac dysfunction, and insulin resistance in obese animal models [52–54]. Mechanistically, OXTRs in pancreatic islets mediate glucose-induced insulin secretion through increased intracellular calcium in rodents, and OT further supports β-cell survival [55,56].

OT significantly attenuated the dyslipidemic effects of OLZ in this study. OLZ has been shown to rapidly alter lipid metabolism through mechanisms which are probably independent of insulin signaling [55]. In healthy volunteers, a short course of OLZ (10 mg/day for 3 days) increased circulating leptin and triglyceride levels [48]. At the molecular level, OLZ upregulates SREBP-2, the master regulator of cholesterol biosynthesis [57]. It also enhances hepatic expression of proprotein convertase subtilisin/kexin type 9 (PCSK9), which degrades LDL receptors, thereby impairing hepatic LDL-C

clearance and raising plasma LDL-C levels [58,59]. Regarding the effect of OLZ on HDL-C, a few studies [60] found that OLZ does not alter HDL-C levels, while several others demonstrated a decrease in HDL-C [61,62]. Further, OLZ-induced insulin resistance is expected to further exacerbate dyslipidemia.

OT, in contrast, possesses antidyslipidemic activity, likely mediated directly or indirectly through enhanced insulin release and improved target tissue sensitivity. OT increases levels of the endogenous lipid oleoylethanolamide, which promotes fatty acid oxidation via activation of PPARα, a key regulator of lipid β-oxidation [63,64]. Therefore, an increase in OT will lead to an increase in fat breakdown and uptake [65]. Moreover, OT has been found to decrease blood triglyceride levels while subsequently increasing blood glycerol levels, supporting its lipid catabolic and absorptive action [66].

While lifestyle modifications remain the preferred initial intervention for managing OLZ-induced metabolic disturbances, their efficacy is often compromised by poor adherence and the drug's direct appetite-stimulating effects [67,68]. In this study, metformin, a biguanide anti-diabetic drug [69], was used as the positive control since it has previously been found to ameliorate OLZ-induced metabolic syndrome [70–72]. However, OT may offer a more targeted therapeutic approach due to its central actions compared to both lifestyle changes and metformin. Emerging evidence demonstrates that OT directly modulates hypothalamic pathways that regulate energy homeostasis and reward centers, thereby reducing OLZ-induced metabolic dysfunction in animal models [66]. This central mechanism suggests potential advantages in pre-venting early neuroendocrine changes, unlike metformin, which mainly acts on peripheral insulin sensitization [67].

Although further research is needed to confirm its relative accuracy and clinical practicality, our findings highlight its potential as a mechanistically distinct and possibly complementary strategy.

There were several limitations of this study. It was conducted in normal rats rather than in schizophrenia rats and therefore the study results cannot be regarded as being 100% translatable to the human clinical setting. However, many previous studies investigating the metabolic effects of OLZ have used the same model as the one used in this study. The exclusive use of female rats decreases inconsistency but prevents the assessment of possible sex-specific differences in metabolic and oxytocinergic responses. We also did not perform molecular assays to confirm specific signaling pathways. Additionally, we did not evaluate oxytocin pharmacokinetics, which previous studies show are characterized by rapid peaks in plasma and the brain around 30 minutes after administration, remaining elevated for about an hour [73]. Future research using neuropsychiatric disease models, both sexes, molecular analyses, and pharmacokinetic profiling would help to strengthen and extend these findings.

## Conclusion

In this study, OT significantly attenuated OLZ-induced metabolic dysfunction. These findings suggest that oxytocinergic hypofunction may contribute to the pathophysiology of OLZ-induced metabolic disturbances and that modulation of this system could represent a promising therapeutic strategy. However, these results are based on preclinical models; further research is needed to confirm the mechanisms, assess translational relevance, and their potential use in clinical settings.

## Supporting information

**S1 File. List of abbreviations.**
(PDF)

**S2 File. Mean food intake during the induction phase.**
(PDF)

**S3 File. Mean body weight during the induction phase.**
(PDF)

**S4 File. Mean random blood glucose during the induction phase.**
(PDF)

**S5 File.  Oral glucose tolerance test during the induction phase.**
(PDF)

**S6 File.  Mean food intake during the treatment phase.**
(PDF)

**S7 File.  Mean body weight during the treatment phase.**
(PDF)

**S8 File.  Mean random blood glucose during the treatment phase.**
(PDF)

**S9 File.  Oral glucose tolerance test during the treatment phase.**
(PDF)

**S10 File.  Oral glucose tolerance test at the end of the treatment phase.**
(PDF)

**S11 File.  Liver weight at the end of the treatment phase.**
(PDF)

**S12 File.  Hepatic index at the end of the treatment phase.**
(PDF)

**S13 File.  Hepatic triglyceride levels at the end of the treatment phase.**
(PDF)

**S14 File.  Hepatic steatosis area at the end of the treatment phase.**
(PDF)

**S15 File.  Mesenteric adipose tissue weight at the end of the treatment phase.**
(PDF)

**S16 File.  Retroperitoneal adipose tissue weight at the end of the treatment phase.**
(PDF)

**S17 File.  Pericardial adipose tissue weight at the end of the treatment phase.**
(PDF)

**S18 File.  Total visceral adipose tissue weight at the end of the treatment phase.**
(PDF)

**S19 File.  Total serum cholesterol at the end of the treatment phase.**
(PDF)

**S20 File.  Serum triglyceride levels at the end of the treatment phase.**
(PDF)

**S21 File.  LDL-C at the end of the treatment phase.**
(PDF)

**S22 File.  HDL-C at the end of the treatment phase.**
(PDF)

**S1 Table. Mean body weight during induction summary table.**
(PDF)

**S2 Table. Mean body weight during treatment summary table.**
(PDF)

**S3 Table. Lipid profile summary table.**
(PDF)

## Acknowledgments

The authors are grateful for the technical assistance that was given by Ms. Carolyne, Dr. Ann Moraa and Mr. Horo Mwaura.

## Author contributions

**Conceptualization:** Elsie D. Oduor, Peter W. Mwangi, Boniface M. Chege, Sharon F. Olago, Frederick Bukachi.

**Data curation:** Elsie D. Oduor, Peter W. Mwangi, Boniface M. Chege, Sharon F. Olago, Frederick Bukachi.

**Formal analysis:** Elsie D. Oduor.

**Investigation:** Elsie D. Oduor, Peter W. Mwangi, Boniface M. Chege, Sharon F. Olago.

**Methodology:** Elsie D. Oduor, Boniface M. Chege, Frederick Bukachi.

**Resources:** Elsie D. Oduor.

**Supervision:** Elsie D. Oduor, Peter W. Mwangi.

**Validation:** Elsie D. Oduor, Boniface M. Chege, Sharon F. Olago, Frederick Bukachi.

**Writing – original draft:** Elsie D. Oduor, Peter W. Mwangi, Sharon F. Olago.

**Writing – review & editing:** Elsie D. Oduor, Peter W. Mwangi, Boniface M. Chege, Sharon F. Olago, Frederick Bukachi.

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
