## [Decision Letter · Decision Letter 0]

24 Jul 2025

PONE-D-25-09455The olanzapine-induced metabolic syndrome appears to be partially caused by the dysfunction of the oxytocinergic system.PLOS ONE

Dear Dr. Oduor,

Thank you for submitting your manuscript to PLOS ONE. After careful consideration, we feel that it has merit but does not fully meet PLOS ONE’s publication criteria as it currently stands. Therefore, we invite you to submit a revised version of the manuscript that addresses the points raised during the review process.

We look forward to receiving your revised manuscript.

Kind regards,

Zahra Lorigooini

Academic Editor

PLOS ONE

Journal Requirements:

2. We note that your Data Availability Statement is currently as follows: 

“All relevant data are within the manuscript and its Supporting Information files”

4. Please upload a new copy of Figures 5 to 8 as the detail is not clear. Please follow the link for more information: 

https://blogs.plos.org/plos/2019/06/looking-good-tips-for-creating-your-plos-figures-graphics/

https://blogs.plos.org/plos/2019/06/looking-good-tips-for-creating-your-plos-figures-graphics/

**Additional Editor Comments:**

**Title and Abstract**:The title is clear but overly cautious with the phrase “appears to be partially caused.” Consider revising to a more assertive and specific title, such as: *“Olanzapine-Induced Metabolic Syndrome Is Partially Mediated by Oxytocinergic System Dysfunction in Female Sprague-Dawley Rats”*.The abstract lacks sufficient methodological details (e.g., number of rats, doses, study duration) and specific results (e.g., effect sizes or numerical values). Revise the abstract to include these details, ensure it is concise (≤300 words), and strengthen the conclusion by linking findings to potential clinical implications or future research directions.**Ethics and Data Availability Statements**:The ethics statement is incomplete, lacking details on the approving ethics committee, approval number, and adherence to animal welfare guidelines. Please provide a comprehensive statement, including the committee name and approval number.The data availability statement is vague and does not meet PLOS ONE’s open data policy. Upload data to a public repository or provide a clear justification for restricted access, including contact details for data requests (e.g., “Data are available from the University of Nairobi Institutional Data Access Committee [contact email]”).**Data Presentation**:The results section is repetitive and lacks structure, making it difficult to follow trends across groups. Organize results into clear subsections (e.g., induction phase outcomes, treatment phase outcomes, inter-group comparisons) and include summary tables or figures to consolidate key findings (e.g., body weight, food intake, lipid profiles).Ensure figures meet PLOS ONE’s technical requirements (≥300 dpi, TIFF/EPS format) and include detailed legends explaining content and statistical significance.**Introduction and Discussion**:The introduction is overly lengthy and lacks focus on the oxytocinergic system’s role. Streamline it into three sections: background on olanzapine and metabolic syndrome, oxytocin’s role in metabolism, and study objectives with a clear hypothesis.The discussion is repetitive and includes excessive mechanistic detail. Shorten it, focusing on key findings, and add a subsection addressing study limitations (e.g., use of female rats, short-term treatment, animal model constraints). Strengthen clinical implications to highlight the study’s translational potential.**Language and Style**:The manuscript contains numerous typographical and grammatical errors (e.g., “titd” instead of “titled,” “mctabolic syndapars” instead of “metabolic syndrome”). Engage a professional editor or use tools like Grammarly to correct these errors.Replace vague terms (e.g., “notable variations”) with precise scientific language (e.g., “statistically significant differences”) and break long sentences into shorter, clearer ones to improve readability.

**Additional Recommendations**

**References**: Update references to prioritize recent studies (2015–2025) and ensure completeness in Vancouver style. **Methodological Details**: Clarify group allocations in the treatment phase, justify chosen doses (e.g., oxytocin, metformin), and provide detailed protocols for assays (e.g., Butler and Mailing for hepatic triglycerides). Specify statistical methods, including software and correction methods. **Visualizations**: Consider adding a figure (e.g., a line graph showing body weight changes over time) to enhance data clarity, as suggested in the detailed review comments.

Reviewers' comments:

Reviewer's Responses to Questions

**Comments to the Author**

1. Is the manuscript technically sound, and do the data support the conclusions?

Reviewer #1: Partly

Reviewer #2: No

Reviewer #3: Yes

2. Has the statistical analysis been performed appropriately and rigorously? 

Reviewer #1: N/A

Reviewer #2: Yes

Reviewer #3: Yes

3. Have the authors made all data underlying the findings in their manuscript fully available?

Reviewer #1: Yes

Reviewer #2: Yes

Reviewer #3: Yes

4. Is the manuscript presented in an intelligible fashion and written in standard English?

Reviewer #1: Yes

Reviewer #2: Yes

Reviewer #3: No

5. Review Comments to the Author

Reviewer #1: The Introduction section is uncompleted and needs a key revise. The authors should explain more about the oxytocinergic system and its possible correlation with OLZ.

What is the OT? Please add its full name before mentioning the OT for the first time in the text and also in the abstract section.

Did the Metformin hydrochloride tablets weigh was 500 grams?!!

which organ was chosen for blood drawing, after the rats were anesthetized?

which section of liver tissue was chosen to determine the triglycerides? did this section of liver tissue was same for all rats?

The “O.d” in line 131 should be edited it can be “OD”

All of the figures have low resolution and should be edited. Also, they have not error bar (SEM)!!

It is suggested to perform histological studies of liver tissues.

The discussion section needs a major revise. The authors should discuss their results and the correlation between the results. This section should be re-wright briefly

Reviewer #2: 1. The method lacks strong novelty, as similar derivative blood glucose levels, oral glucose tolerance techniques have already been reported in the literature.

2. The abstract contains detail and should be more concise, focusing on key outcomes and significance.

3. The introduction provides adequate background but does not clearly justify the need for this method over existing Oxytocin, on the other hand, is known to have roles in regulation of feeding behavior, as well as lipid and

26 glucose metabolism.

4. Experimental methodology is detailed, but some steps—such as wavelength selection and validation criteria—need clearer justification.

5. Figures and tables are numerous but not well integrated into the discussion, causing redundancy.

6. The discussion lacks comparison with alternative established methods, particularly in terms of sensitivity, accuracy, and practical advantages In contrast, central OT has been shown to exert anorexigenic effects with direct oxytocinergic neural .

8. The greenness evaluation (NPY/AgRP) is a strong point but lacks a deep critical analysis and clear benchmarking against literature.

9. Language and grammar issues are frequent; the manuscript needs professional editing for clarity and consistency.

10. The conclusion is somewhat overstated and should be more balanced, acknowledging limitations and potential improvements.

11. The references include some outdated or low-impact sources and suffer from inconsistent formatting and integration into the main text.

Reviewer #3: This study looks at whether oxytocin can reduce the negative side effects of olanzapine, a drug used to treat mental illness. Olanzapine can cause weight gain, high blood sugar, and high fat levels, which can lead to metabolic syndrome. The authors tested their idea on female rats. They found that giving oxytocin with olanzapine helped lower food intake, body weight, blood sugar, and fat buildup. The results were compared to both untreated rats and rats given metformin.

Major Strengths

1. Important Topic

Olanzapine is widely used, but its side effects are a real concern. This study tackles a serious problem and explores a possible solution.

2. Clear Design

The groups and treatments are well organized. The study covers many important health markers like body weight, blood sugar, and liver fat.

3. Good Comparisons

Including a metformin group makes the study stronger, since metformin is already known to help with metabolic issues.

4. Logical Discussion

The authors give reasonable explanations for how oxytocin might work in the brain and body to reduce these side effects.

What Needs to Be Improved

1. No Lab Tests for Molecules

The study talks a lot about how oxytocin and olanzapine affect certain pathways in the body. But the absence of molecular analyses (e.g., Western blot for AMPK, SREBP, GLUT4, OXTR expression) limits mechanistic conclusions. The authors should mention this as a limitation.

2. Sex-Specific Limitations

Using only female rats limits the study. The authors say why they chose them, but they should also clearly say that results might differ in males or in humans.

3. Oxytocin Dosage Info Missing

The paper doesn’t explain how long oxytocin stays in the body after injection or how much actually reaches the brain. This should be briefly mentioned.

4. Figures Could Be Better

Make sure all graphs and figure legends are clear and easy to read on their own.

5. Writing Style

Some parts are too formal. A few short edits could make it easier to read.

Minor Suggestions

1. Say whether food intake was measured per rat or per cage.

2. Use the same units throughout the paper (e.g., mmol/L or mg/dL—don’t switch between them).

3. Add a short list of abbreviations.

This study is well done and has practical value. It shows that oxytocin might help reduce side effects caused by olanzapine. With a few edits to add clarity and improve the figures, the paper will be ready for publication. It’s a solid step forward in a field that needs new ideas.

6. PLOS authors have the option to publish the peer review history of their article (what does this mean?). If published, this will include your full peer review and any attached files.

Reviewer #1: No

Reviewer #2: No

Reviewer #3: No

---

## [Author Response · Author response to Decision Letter 1]

5 Sep 2025

To the Editor, PLoS ONE,

The authors would like to thank the journal for the high-quality and professional peer review of our manuscript. The editorial, as well as the peer review comments, have greatly improved the quality of the manuscript. Here are our detailed responses as well as corrections to the editorial and peer review comments.

Journal Requirements

1. Editor: Please ensure that your manuscript meets PLOS ONE's style requirements, including those for file naming. The PLOS ONE style templates can be found at

Authors’ response: The authors’ appreciate the editor’s comment and the manuscript has been revised to meet PLOS ONE requirements.

2. Editor: We note that your Data Availability Statement is currently as follows:

“All relevant data are within the manuscript and its Supporting Information files”

Authors’ response: The authors appreciate the editor’s comment and the authors have uploaded the minimal data as Supporting information files.

3. Editor: PLOS requires an ORCID iD for the corresponding author in Editorial Manager on papers submitted after December 6th, 2016. Please ensure that you have an ORCID iD and that it is validated in Editorial Manager. To do this, go to ‘Update my Information’ (in the upper left-hand corner of the main menu), and click on the Fetch/Validate link next to the ORCID field. This will take you to the ORCID site and allow you to create a new iD or authenticate a pre-existing iD in Editorial Manager.

Authors’ response: We thank the editor for this reminder. The corresponding author’s ORCID iD has been added and successfully validated in Editorial Manager.

4. Please upload a new copy of Figures 5 to 8 as the detail is not clear. Please follow the link for more information:

https://blogs.plos.org/plos/2019/06/looking-good-tips-for-creating-your-plos-figures-graphics/

https://blogs.plos.org/plos/2019/06/looking-good-tips-for-creating-your-plos-figures-graphics/

Authors’ response: We thank the editor for the comment. All figures have been revised to follow PLOS ONE guidelines, and the Results section has been restructured as suggested. As a result, the manuscript now includes 13 figures, all formatted according to the journal’s requirements.

EDITOR AND REVIEWERS COMMENTS:

1. Editor: The title is clear but overly cautious with the phrase “appears to be partially caused.” Consider revising to a more assertive and specific title, such as: “Olanzapine-Induced Metabolic Syndrome Is Partially Mediated by Oxytocinergic System Dysfunction in Female Sprague-Dawley Rats”.

Authors’ response: We thank the reviewer for this comment. We have revised it as suggested.

2. Editor: The abstract lacks sufficient methodological details (e.g., number of rats, doses, and study duration) and specific results (e.g., effect sizes or numerical values). Revise the abstract to include these details, ensure it is concise (≤300 words), and strengthen the conclusion by linking findings to potential clinical implications or future research directions.

Authors’ response: The authors acknowledge the editor’s comment. We have added methodological detail: number of rats, doses and study duration, and specific results in the abstract. We have revised the conclusion in the abstract to explicitly link our findings to clinical implications and future research directions. The revised text now reads: “OLZ-induced metabolic abnormalities were mitigated by oxytocin, indicating that the oxytocinergic system hypofunction may be implicated in its pathophysiology. These results highlight OT’s therapeutic potential and call for further clinical research to explore its role in the management of antipsychotic-induced metabolic syndrome.” We believe this revision clearly conveys both the potential clinical relevance and the need for future studies.

3. Editor: The ethics statement is incomplete, lacking details on the approving ethics committee, approval number, and adherence to animal welfare guidelines. Please provide a comprehensive statement, including the committee name and approval number.

Authors’ response: We thank the editor for this observation. We would like to respectfully clarify that the ethics statement, including the approving ethics committee, approval number, and adherence to animal welfare guidelines, is currently included in the manuscript. The ethics statement in the manuscript is as follows: “Ethical approval for the experimental study was obtained from the Biosafety, Animal Use and Ethics Committee, Department of Veterinary Anatomy and Physiology, Faculty of Veterinary Medicine, University of Nairobi, and FVM BAUEC/2023/440 was the permit number issued. All surgical procedures were carried out under sodium pentobarbital anesthesia, with rigorous measures taken to reduce distress and discomfort.”

4. Editor: The data availability statement is vague and does not meet PLOS ONE’s open data policy. Upload data to a public repository or provide a clear justification for restricted access, including contact details for data requests (e.g., “Data are available from the University of Nairobi Institutional Data Access Committee [contact email]”).

Authors’ response: We thank the editor for this comment. All relevant data are included in the Supporting Information files, which are publicly accessible with the manuscript.

5. Editor: The results section is repetitive and lacks structure, making it difficult to follow trends across groups. Organize results into clear subsections (e.g., induction phase outcomes, treatment phase outcomes, inter-group comparisons) and include summary tables or figures to consolidate key findings (e.g., body weight, food intake, lipid profiles).

Authors’ response: We appreciate the editor’s insightful suggestion. We have restructured the Results section into clear subsections and added figures to consolidate key findings. We have included summary tables under Supporting Information files (S24 Table, S23 Table and S25 Table).

6. Editor: Ensure figures meet PLOS ONE’s technical requirements (≥300 dpi, TIFF/EPS format) and include detailed legends explaining content and statistical significance.

Authors’ response: We appreciate the editor’s guidance regarding figure quality and presentation. All figures have been revised to comply with PLOS ONE’s technical requirements and included detailed legends and captions explaining content and statistical significance.

7. Editor: The introduction is overly lengthy and lacks focus on the oxytocinergic system’s role. Streamline it into three sections: background on olanzapine and metabolic syndrome, oxytocin’s role in metabolism, and study objectives with a clear hypothesis.

Author’s response: We sincerely thank the editor for this valuable feedback. In line with the recommendation, we have revised and streamlined the Introduction into three focused sections as suggested. We believe this restructuring improves the clarity, flow, and focus of the Introduction.

Paragraph 1: background on olanzapine and metabolic syndrome

“Olanzapine (OLZ) is an atypical antipsychotic drug used in the management of psychiatric disorders such as schizophrenia, depression, and bipolar disorder (1). Long-term OLZ therapy is associated with metabolic syndrome as a side effect (2,3), a complex of metabolic dysregulations including central obesity, dyslipidemia, hyperglycemia, and hypertension (3). Metabolic syndrome is linked to cardiovascular disease, type 2 diabetes mellitus, and overall increased mortality (4). However, the pathophysiologic mechanisms underlying olanzapine-induced metabolic dysregulation remain unclear. Proposed contributing factors include the disruption of appetite-regulating centers, reduced peripheral insulin sensitivity, and altered lipid metabolism (5).”

Paragrapg 2: Oxytocin’s role in metabolism

“Beyond its well-established role in reproduction, oxytocin has been found to influence feeding behavior and energy metabolism, with regard to glucose and lipid metabolism (6). Oxytocin receptors (OXTR) are present in several organs involved in energy metabolism and utilization, e.g., adipose tissue, pancreas, liver, skeletal muscle, and the hypothalamus (7). A deficiency in oxytocin signaling is associated with insulin resistance, weight gain, and dyslipidemia (8). Unlike other known intervention methods like metformin, which mainly improve insulin sensitivity and decrease hepatic glucose production(9), oxytocin has been shown to regulate metabolism through both central mechanisms, such as appetite and energy balance, and peripheral actions in metabolic organs (7). However, its involvement in olanzapine-induced metabolic syndrome is yet to be fully elucidated.”

Paragraph 3: Study objectives with a clear hypothesis

“This study aimed to explore the involvement of the oxytocinergic system as a possible pathophysiologic mechanism underlying OLZ-induced metabolic syndrome in a rodent model by, analyzing dose-dependent effects (OLZ), characterizing its biochemical features, and investigating OT’s therapeutic potential. We hypothesized that OLZ-induced metabolic syndrome is partially mediated by hypofunction of the oxytocinergic system.”

8. Editor: The discussion is repetitive and includes excessive mechanistic detail. Shorten it, focusing on key findings, and add a subsection addressing study limitations (e.g., use of female rats, short-term treatment, animal model constraints). Strengthen clinical implications to highlight the study’s translational potential.

Authors’ response: We thank the editor for this constructive feedback. The Discussion has been revised to reduce repetition and excessive mechanistic detail. We have shortened it, with greater emphasis placed on the key findings. We have also added a dedicated subsection on study limitations. We have strengthened the clinical implications to highlight the study’s translational potential.

9. Editor: The manuscript contains numerous typographical and grammatical errors (e.g., “titd” instead of “titled,” “mctabolic syndapars” instead of “metabolic syndrome”). Engage a professional editor or use tools like Grammarly to correct these errors.

Authors; response: We thank the editor for highlighting these issues. The manuscript has been carefully reviewed and corrected for typographical and grammatical errors throughout. We have used both manual proofreading and grammar-checking tools to ensure accuracy and readability.

10. Editor: Replace vague terms (e.g., “notable variations”) with precise scientific language (e.g., “statistically significant differences”) and break long sentences into shorter, clearer ones to improve readability.

Authors’ response: We have revised the manuscript to replace vague terms with precise scientific language. In addition, we have broken down long sentences into shorter, clearer ones to improve readability.

11. Editor: References: Update references to prioritize recent studies (2015–2025) and ensure completeness in Vancouver style

Authors’ response: The authors’ thank the editor for this suggestion.We have revised the reference list to prioritize recent literature (2015–2025) and to ensure full compliance with Vancouver style. Only a very small number of older citations were retained, as they are important guideline or background sources needed for context, for example, to justify doses used in the methodology and to support standard experimental procedures.

12. Editor: Methodological Details: Clarify group allocations in the treatment phase, justify chosen doses (e.g., oxytocin, metformin), and provide detailed protocols for assays (e.g., Butler and Mailing for hepatic triglycerides). Specify statistical methods, including software and correction methods.

Authors’ response: We appreciate the constructive comments.

The manuscript provides a detailed description of group allocations. To further enhance clarity, we have added a figure illustrating the group allocations.

The doses of oxytocin and metformin were selected based on previous literature. Oxytocin was adopted from an anti-obesity study, while metformin was derived from a study on olanzapine-induced metabolic syndrome, ensuring efficacy and relevance in this model. We have included these details in the Methods section and referenced accordingly.

Citation for the oxytocin dose:

Birech Z, Mwangi P, Sehmi P, Nyaga N. Application of Raman spectroscopy in comparative study of antiobesity influence of oxytocin and freeze‐dried extracts of Uvariodendron anisatum Verdeck ( Annonaceae ) in Sprague Dawley rats. J Raman Spectrosc. 2019 Nov 25;51.

Citation for metformin dose:

Luo C, Wang X, Huang HX, Mao XY, Zhou HH, Liu ZQ. Coadministration of metformin prevents olanzapine-induced metabolic dysfunction and regulates the gut-liver axis in rats. Psychopharmacology (Berl). 2021 Jan 1;238(1):239–48.

We have provided detailed protocols for all assays, including the Butler and Maling method for hepatic triglyceride estimation, in the Methods section. We have provided sufficient detail to ensure reproducibility, while also citing the original publication for reference.

The statistical methods have been specified in the revised manuscript, including the software used (GraphPad Prism 8.0.2) and the correction method (Tukey’s post hoc test for multiple comparisons).

13. Editor: Visualizations: Consider adding a figure (e.g., a line graph showing body weight changes over time) to enhance data clarity, as suggested in the detailed review comments.

Authors’ response: We appreciate the editor’s suggestion and have added 12 figures to enhance data clarity.

Reviewer #1:

1. Reviewer: The Introduction section is uncompleted and needs a key revise. The authors should explain more about the oxytocinergic system and its possible correlation with OLZ.

Authors’ response: We thank the reviewer for this valuable comment. In response, we have revised and expanded the Introduction to provide a more detailed explanation of the oxytocinergic system and its possible correlation with olanzapine.

The Introduction has been updated with the following paragraph: “Beyond its well-established role in reproduction, oxytocin has been found to influence feeding behavior and energy metabolism, with regard to glucose and lipid metabolism (6). Oxytocin receptors (OXTR) are present in several organs involved in energy metabolism and utilization, e.g., adipose tissue, pancreas, liver, skeletal muscle, and the hypothalamus (7). A deficiency in oxytocin signaling

---

## [Decision Letter · Decision Letter 1]

6 Oct 2025

The olanzapine-induced metabolic syndrome appears to be partially caused by the dysfunction of the oxytocinergic system.

PONE-D-25-09455R1

Dear Dr. Oduor,

We’re pleased to inform you that your manuscript has been judged scientifically suitable for publication and will be formally accepted for publication once it meets all outstanding technical requirements.

Kind regards,

Jianhong Zhou

Staff Editor

PLOS ONE

Additional Editor Comments (optional):

Reviewers' comments:

Reviewer's Responses to Questions

**Comments to the Author**

1. If the authors have adequately addressed your comments raised in a previous round of review and you feel that this manuscript is now acceptable for publication, you may indicate that here to bypass the “Comments to the Author” section, enter your conflict of interest statement in the “Confidential to Editor” section, and submit your "Accept" recommendation.

Reviewer #2: (No Response)

Reviewer #3: All comments have been addressed

2. Is the manuscript technically sound, and do the data support the conclusions?

Reviewer #2: (No Response)

Reviewer #3: Yes

3. Has the statistical analysis been performed appropriately and rigorously? 

Reviewer #2: I Don't Know

Reviewer #3: Yes

4. Have the authors made all data underlying the findings in their manuscript fully available?

Reviewer #2: No

Reviewer #3: Yes

5. Is the manuscript presented in an intelligible fashion and written in standard English?

Reviewer #2: Yes

Reviewer #3: Yes

6. Review Comments to the Author

Reviewer #2: Overall Recommendation I recommend the acceptance of this manuscript, contingent upon addressing the revisions suggested above.

Reviewer #3: The study tests whether oxytocin can reduce the harmful metabolic effects of olanzapine in female rats. The revised version provides a lot of detail on the design, results, and possible mechanisms. Overall, the manuscript is stronger and clearer than the first version and the results are convincing.

7. PLOS authors have the option to publish the peer review history of their article (what does this mean?). If published, this will include your full peer review and any attached files.

Reviewer #2: No

Reviewer #3: No

---

## [Editor Report · Acceptance letter]

PONE-D-25-09455R1

PLOS ONE

Dear Dr. Oduor,

I'm pleased to inform you that your manuscript has been deemed suitable for publication in PLOS ONE. Congratulations! Your manuscript is now being handed over to our production team.

Kind regards,

on behalf of

Dr. Jianhong Zhou

Staff Editor

PLOS ONE